



**How does water yield respond to mountain pine beetle infestation in a**
**semiarid forest?**
Jianning Ren[1,3], Jennifer Adam[1], Jeffrey A. Hicke[2], Erin Hanan[3], Naomi Tague[4], Mingliang Liu[1],
Crystal Kolden[5], John T. Abatzoglou[5]
[1] Department of Civil & Environmental Engineering, Washington State University, 99163,
Pullman, USA
[2] Department of Geography, University of Idaho, 83844, Moscow, USA
[3] Department of Natural Resources and Environmental Sciences, University of Nevada, 89501,
Reno, USA
[4] Bren School of Environmental Science & Management, University of California, 93106, Santa
Barbara, USA
[5] Management of Complex Systems, University of California, 95344, Merced, USA
*Corresponding to: Jennifer Adam (jcadam@wsu.edu)*



**Key points:**
• Mountain pine beetle (MPB)-caused tree mortality increases water yield in most wet
years, and a decrease in water yield mainly happens in dry years; therefore, interannual
climate variability is an important driver of water yield response to beetle-caused tree
mortality.
• A long-term (multi-decade) aridity index is a reliable indicator of water yield response to
MPBs: in a dry year, decreases occur mainly in "water-limited" areas and vegetation
mortality levels have only minor effects; in wetter areas, decreases only occur at low
mortality levels.
• Generally, in a dry year, low to medium MPB-caused vegetation mortality decreases
water yield, and high mortality increases water yield; this response to mortality level is
nonlinear and varies by location and year.



**Graphical abstract**

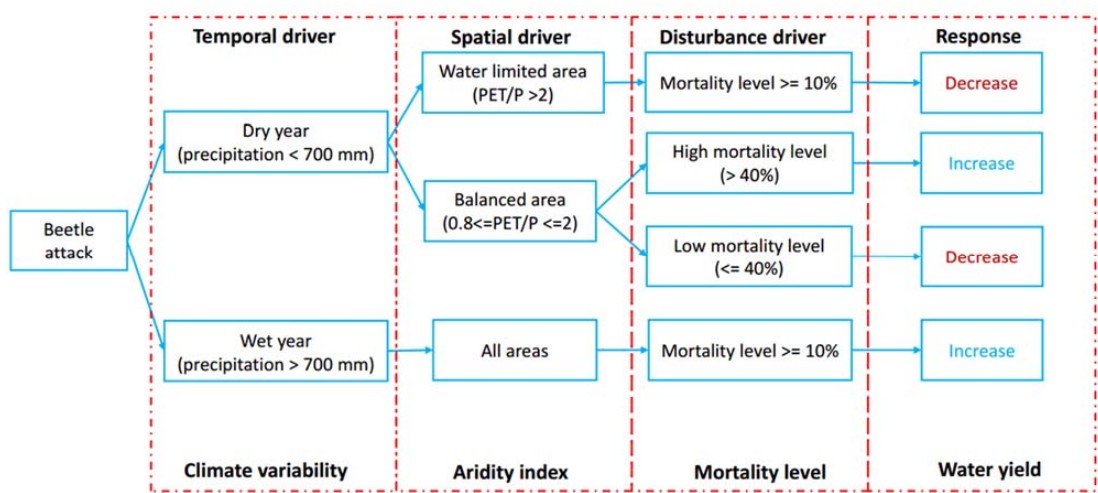

**Abstract**
Mountain pine beetle (MPB) outbreaks in western United States result in widespread tree
mortality, transforming forest structure within watersheds. While there is evidence that these
changes can alter the timing and quantity of streamflow, there is substantial variation in both the
magnitude and direction of responses and the climatic and environmental mechanisms driving
this variation are not well understood. Herein, we coupled an eco-hydrologic model (RHESSys)
with a beetle effects model and applied it to a semiarid watershed, Trail Creek, in the Bigwood
River basin in central Idaho to evaluate how varying degrees of beetle-caused tree mortality
influence water yield. Simulation results show that water yield during the first 15 years after
beetle outbreak is controlled by interactions among interannual climate variability, the extent of
vegetation mortality, and long-term aridity. During wet years, water yield after beetle outbreak
increases with greater tree mortality. During dry years, water yield decreases at low to medium
mortality but increases at high mortality. The mortality threshold for the direction of change is



location-specific. The change in water yield also varies spatially along aridity gradients during
dry years. In relatively wetter areas of the Trail Creek basin, water yield switches from a
decrease to an increase when vegetation mortality is greater than 40 percent. In more water-
limited areas on the other hand, water yield typically decreases after beetle outbreaks, regardless
of mortality level. Results suggest that long-term aridity can be a useful indicator for the
direction of water yield changes after disturbance.
**1 Introduction**
In recent decades, mountain pine beetle (MPB) outbreaks in the Western U.S. and Canada have
killed billions of coniferous trees (Bentz et al. 2010). Coniferous forests can provide essential
ecosystem services, including water supply for local communities (Anderegg et al. 2013).
Therefore, it is essential to understand how ecosystems and watersheds respond to beetle
outbreaks and to identify the dominant processes that drive these responses (Bennett et al. 2018).
A growing number of studies have qualitatively examined hydrologic responses to beetle
outbreaks and disturbance; however these studies have produced conflicting results (Adams et al.
2012; Goeking and Tarboton 2020). While some studies show increases in water yield following
beetle outbreak (e.g., Bethlahmy 1974; Potts 1984; Livneh et al. 2015), many others show no
change or even decreases (e.g., Guardiola-Claramonte et al. 2011; Biederman et al. 2014; Slinski
et al. 2016). To determine which mechanisms control change in water yield following beetle
outbreak, more quantitative approaches are needed.
Water yield is often thought to increase after vegetation is killed or removed by disturbances
such as fire, thinning, and harvesting (Hubbart 2007; Robles et al. 2014; Chen et al. 2014; Buma
and Livneh 2017; Wine et al. 2018). In the Rocky Mountain West, beetle outbreaks have
increased water yield through multiple mechanisms. First, defoliation/needle loss can reduce



plant transpiration, canopy evaporation, and canopy snow sublimation losses to the atmosphere
(Montesi et al. 2004). Increased canopy openings can also enable snow accumulation and allow
more radiation to reach the ground surface, resulting in earlier and larger peak snowmelt events,
which can in turn reduce soil moisture and therefore decrease summer evapotranspiration (ET).
Several studies have documented decreases in water yield following disturbances (e.g., mortality,
fire, beetle outbreaks; Biederman et al. 2014; Bart et al. 2016; Slinski et al. 2016; Goeking and
Tarboton 2020). For example, in the southwestern U.S., beetle outbreaks have decreased
streamflow by opening forest canopies and increasing radiation to the understory and at the
ground surface, which leads to increases in understory vegetation transpiration (Guardiola-
Claramonte et al. 2011), soil evaporation, and therefore increases total ET (Bennett et al. 2018).
Tree  mortality or removal can reduce streamflow because surviving trees and/or understory
vegetation compensates by using more water (Tague et al. 2019).
In a review of 78 studies, Goeking and Tarboton (2020) concluded that the decrease in water
yield after tree-mortality mainly happens in semiarid regions. Previous studies also provide rule-
of-thumb thresholds above which water yield will increase: at least 20 percent loss of vegetation
cover and mean precipitation of 500 mm/year (Adams et al. 2012). However, many watersheds
in the western U.S. experience high interannual climate variability (Fyfe et al. 2017), and local
environmental gradients (e.g., long-term aridity gradients) may strongly influence vegetation and
hydrologic responses to disturbances, including beetle outbreaks, making predictions difficult
(Winkler et al. 2014). Given the possibility of either increases or decreases in water yield
following beetle outbreaks, modeling approaches are crucial for identifying the specific
mechanisms that control these responses.



The overarching goal of this study is to identify mechanisms driving the direction of change in annual water yield after beetle outbreaks in semi-arid regions (note that in the following text, "water yield" refers to means annual water yield). The following specific questions address this goal:

- **Q1:** What is the role of **interannual climate variability** in water yield response?
- **Q2:** What is the role of **mortality level** in water yield response?
- **Q3:** How does **long-term aridity** (defined as temporally averaged potential evapotranspiration relative to precipitation for a period of 38 years) modify these responses, and how do responses vary spatially within a watershed along aridity gradients?

We hypothesize that multiple ecohydrologic processes (e.g., snow accumulation and melt, evaporation, transpiration, drainage, and a range of forest structural and functional responses to beetles) could interactively influence how water yield responds to beetle outbreaks—however, in certain locations one or more processes may dominate. In addition, the dominant ecohydrologic processes may vary over space and time due to interannual climate variability (i.e., precipitation), vegetation mortality, and long-term aridity. In Sect 2, we present a conceptual framework for identifying and depicting dominant hydrological processes through which forests respond to beetle infestation. We use this framework to interpret the modeling results. In Sect 3, we describe our mechanistic modeling approach, i.e., using the Regional Hydro-Ecological Simulation System (RHESSys), which can prescribe a range of vegetation mortality levels, capture the effects of landscape heterogeneity and the role of lateral soil moisture redistribution, and project ecosystem carbon and nitrogen dynamics, including post-disturbance plant recovery.



In Sects 4 and 5, we then present modeling results that explore how multiple mechanisms
influence water yield responses.
**2 Conceptual framework**
2.1 Vegetation response to beetle outbreaks
Mountain pine beetles (MPB) introduce blue stain fungi into the xylem of attacked trees, which
reduces water transport in plants and eventually shuts it off (Paine et al. 1997). During outbreaks,
MPBs prefer to attack and kill larger host trees that have greater resources (e.g., carbon), while
smaller diameter host trees and non-host vegetation (including the understory) remain unaffected
(Edburg et al. 2012). After MBP outbreak, trees mainly go through three phases (i.e., red, gray,
and old) over time (Hicke et al. 2012). During the red phase, the trees' needles turn red. During
the gray phase, there are no needles in the canopy. During old phase, killed trees have fallen, and
understory vegetation and new seedlings experience rapid growth (Hicke et al. 2012; Mikkelson
et al. 2013).
2.2 Hydrologic response to beetle outbreaks
Figure 1 describes the main processes that alter evapotranspiration to either decrease or increase
water yield, depending on which processes dominate (Adams et al. 2012; Goeking and Tarboton
2020). During the red and gray phases, needles fall to the ground, and there is lower leaf area
index (LAI) and a more open canopy (Hicke et al. 2012). This can reduce plant transpiration of
infected trees, though remaining trees may compensate to some extent by increasing
transpiration in water limited environments (Adams et al. 2012, Tague et al. 2019). A more open
canopy intercepts less precipitation, reducing evaporation from the canopy but potentially
increasing it from soil and litter layers (Montesi et al. 2004; Sexstone et al. 2018). Meanwhile, an
open canopy can increase the proportion of snow falling to the ground and, therefore, increase



snowpack accumulation. With more solar radiation reaching the ground, earlier and larger peak
snowmelt can also occur (Bennett et al. 2018). Generally, earlier snowmelt increases water for
spring streamflow and decreases water for summertime ET (Pomeroy et al. 2012). However,
once snags fall, reductions in longwave radiation can actually lead to later snowmelt (Lundquist
et al. 2013). The open canopy and less competition for resources, such as solar radiation and
nutrients, can also promote understory vegetation growth, which may increase understory
transpiration  (Biederman et al. 2014; Tague et al. 2019). Whether water yield increases or
decreases will ultimately depend on the balance of these processes that can alter transpiration and
evaporation in different ways.

Finally, interannual variability in climate (e.g., dry versus wet years) can affect forests'
hydrological responses (Winkler et al. 2014; Goeking and Tarboton 2020). For instance, during
wet years, remaining plants are not water-limited, and reductions in plant transpiration due to
beetle-caused mortality dominate increases in soil evaporation or remaining plant transpiration,
resulting in a higher water yield.  In contrast, during dry years, plants are already under water
stress and decreases in plant transpiration caused by tree mortality may be compensated by
increasing soil evaporation and transpiration by remaining trees or understory vegetation, leading
to declines in water yield. Moreover, these responses are also affected by land cover types (e.g.,
young vs old pine, different tree species, etc.), which is not currently well documented (Perry
and Jones 2017; Morillas et al. 2017).
2.3 Review of modeling approaches
Many models, ranging from empirical and lumped to physically-based and fully-distributed,
have been used to study hydrologic responses to disturbances. Goeking and Tarboton (2020)





argue that only physically-based and fully-distributed models can capture how disturbances alter
water yield because they represent fine-scale spatial heterogeneity and physical process that vary
over space and time. Despite their advantages, process-based models, such as the coupled CLM-
ParFlow model (Mikkelson et al. 2013; Penn et al. 2016), the Distributed Hydrology Soil
Vegetation Model (Livneh et al. 2015; Sun et al. 2018), and the Variable Infiltration Capacity
Model (Bennett et al. 2018) also have some limitations. For example, 1) they may assume
constant LAI after disturbances and static vegetation growth (e.g., VIC and DHSVM), 2) they
may not include lateral flow to redistribute soil moisture (VIC), and 3) in some cases, the
approach to represent the effects of beetle outbreaks may be too simplified (e.g., changing only
LAI and conductance without considering two-way beetle-vegetation interactions in post-
disturbance biogeochemical and water cycling e.g., as in CLM-ParFlow). Thus, improving
current fully distributed process-based models to capture the coupled dynamics between
hydrology and vegetation at multiple scales is a critical step for projecting how beetle outbreaks
will affect water yield in semiarid systems (Goeking and Tarboton 2020). Here we use
RHESSys7.1, which captures these processes.
**3 Model, data, and simulation experiment design**
3.1 Study area
Our study watershed is Trail Creek, which is located in Blaine County between the Sawtooth
National Forest and the Salmon-Challis National Forest (43.44N, 114.19W; Fig. 2). It is a 167-
$km^2$ sub-catchment in the south part of Big Wood River basin, and is within the wildland-urban
interface where residents are vulnerable to the flood and debris flows caused by forest
disturbances (Skinner 2013). Trail Creek has frequently experienced beetle outbreaks, notably in
2004 and 2009, when beetles killed 7 and 19 $km^2$ of trees, respectively (Berner et al. 2017).

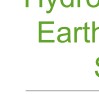 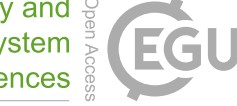

Trail Creek has cold, wet winters and warm, dry summers; mean annual precipitation is
approximately 978 mm with 60% snow (Frenzel 1989). The soil is mostly permeable coarse
alluvium (Smith 1960). Vegetation is clustered into two major groups along the elevation which
ranges from 1760 to 3478 m: sagebrush, riparian species, and grasslands in lower to middle
elevation areas and Douglas-fir (*Pseudotsuga menziesii*), lodgepole pine (*Pinus contorta* var.
*latifolia*), subalpine fir (*Abies lasiocarpa*), and mixed shrub and herbaceous vegetation in
middle-to-higher elevations (Buhidar 2002).
A strong upper to lower vegetation and long-term aridity gradient exists for Trail Creek (Fig. 3).
The northern (higher elevation) portion of the basin is mesic and covered principally by
evergreen forest; the southern (lower elevation) portion is xeric and covered by shrubs, grasses,
and mixed herbaceous species. In total, Trail creek contains 72 sub-basins and two of them (e.g.,
Fig. 3, sub-basin 412 and 416) are urban areas. If we classify this basin into different zones
according to an aridity index, i.e., the ratio of 38-year average annual potential
evapotranspiration (PET) to precipitation (P) (Sect 3.4), there is a distinct gradient: the northern
and high elevation area is balanced (i.e., PET/P between 0.8 and 2) and evergreen tree coverage
is more than 50%; the southern part is water-limited (i.e., PET/P > 2) and evergreen tree
coverage is less than 30% (Figs. 2 and 3).
3.2 Model descriptions
3.2.1 Ecohydrologic model
The Regional Hydro-ecologic Simulation System (RHESSys) (Tague and Band 2004) is a
mechanistic model designed to simulate the effects of climate and land use change on ecosystem
carbon and nitrogen cycling and hydrology. RHESSys fully couples hydrological processes
(including streamflow, lateral flow, ET, and soil moisture, etc.), plant growth and vegetation


dynamics (including photosynthesis, maintenance respiration, and mortality, etc.), and soil
biogeochemical cycling (including soil organic matter decomposition, mineralization,
nitrification, denitrification, and leaching, etc.). It has been widely tested and applied in several
mountainous watersheds in western North America, including many in the Pacific and Inland
Northwest (e.g., Tague and Band 2004; Garcia and Tague 2015; Hanan et al. 2017; Hanan et al.
2018; Lin et al. 2019; Son and Tague 2019).
RHESSys represents a watershed using a hierarchical set of spatial units, including patches,
zones, sub-basins, and the full basin, to simulate various hydrologic and biogeochemical
processes occurring in these multiple scales (Tague and Band 2004). The patch is the finest
spatial scale at which vertical soil moisture and soil biogeochemistry are simulated. In every
patch, there are multiple canopy strata layers to simulate the biogeochemical processes related to
plant growth and nutrient uptake. Meteorological forcing inputs (e.g., temperature, precipitation,
humidity, wind speed, and solar radiation) are handled at the zone level, and spatially
interpolated and downscaled for each patch based on elevation, slope, and aspect. Sub-basins are
closed drainage areas entering both sides of a single stream reach (the water budget is closed in
sub-basins). The largest spatial unit is the basin, which aggregates the streamflow from sub-
basins (Tague and Band 2004; Hanan et al. 2018). In RHESSys, streamflow is the sum of
overland flow and baseflow, and we consider streamflow as the *water yield* of each sub-basin.
RHESSys models vertical and lateral hydrologic fluxes, including canopy interception, plant
transpiration, canopy evaporation/sublimation, snow accumulation, snowmelt and sublimation,
soil evaporation, soil infiltration, and subsurface drainage. Canopy interception is based on the
water-holding capacity of vegetation, which is also a function of plant area index (PAI). Both the
canopy evaporation and transpiration are modeled using the standard Penman-Monteith equation



(Monteith 1965). Snow accumulation is calculated from incoming precipitation and is assumed
to fall evenly across each zone. Snowmelt is based on a quasi-energy budget approach
accounting for radiation input, sensible and latent heat fluxes, and advection. Soil evaporation is
constrained by both energy and atmospheric drivers, as well as a maximum exfiltration rate,
which is controlled by soil moisture (Tague and Band 2004). Vertical drainage and lateral flow is
a function of topography and soil hydraulic conductivity, which decays exponentially with depth
(Tague and Band 2004; Hanan et al. 2018).
Vegetation carbon and nitrogen dynamics are calculated separately for each canopy layer within
each patch, while soil and litter carbon and nitrogen cycling are simulated at the patch level.
Photosynthesis is calculated based on the Farquhar model considering the limitations of nitrogen,
light, stomatal conductance (which is influenced by soil water availability), vapor pressure
deficit, atmospheric $CO_2$ concentration, radiation, and air temperature (Farquhar and von
Caemmerer 1982; Tague and Band 2004). Maintenance respiration is based on Ryan (1991),
which computes respiration as a function of nitrogen concentration and air temperature. Growth
respiration is calculated as a fixed ratio of new carbon allocation for each vegetation component
(Ryan 1991; Tague and Band 2004). Net photosynthesis is allocated to leaves, stems, and roots at
daily steps based on the Dickinson partitioning method, which varies with each plant
development stage (Dickinson et al. 1998). LAI is estimated from leaf carbon and specific leaf
area for each vegetation type. The soil and litter carbon and nitrogen cycling (heterotrophic
respiration, mineralization, nitrification, and denitrification, etc.) are modified from the
BIOME_BGC and CENTURY-NGAS models (White and Running 1994; Parton et al. 1996;
Tague and Band 2004). A detailed description of RHESSys model algorithms can be found in
Tague and Band (2004).



### 3.2.2 Beetle effects model

Edburg et al. (2012) designed and developed a model of MPB effects on carbon and nitrogen

dynamics for integration with the Community Land Model Version 4 (CLM4) (Lawrence et al.

2011, Fig. 4). Here we integrated this beetle effects model into RHESSys (Fig. 4). Beetles attack

trees mainly during late summer, and needles will turn from green to red at the beginning of the

following summer. We simplify this process with prescribed tree mortality on September 1 to

represent a beetle outbreak of the current year. The advantage of this integration is that RHESSys

accounts for the lateral connectivity in water and nitrogen fluxes among patches which is not

represented in CLM4 (Fan et al. 2019). Differences in our approach compared to other

hydrological models of beetle effects (e.g., VIC, CLM-ParFlow, and DHSVM) include dynamic

changes in plant carbon and nitrogen cycling caused by beetle attack, plant recovery, and effects

on hydrological responses. Previous studies of hydrologic effects of beetle outbreaks have

mainly focused on consequences of changes in LAI and stomatal resistance during each phase of

beetle outbreak but have missed feedbacks between carbon and nitrogen dynamics, vegetation

recovery, and hydrology (Mikkelson et al. 2013; Livneh et al. 2015; Penn et al. 2016; Sun et al.

2018; Bennett et al. 2018).

To better represent the effects of beetle-caused tree mortality, we added a snag pool (standing

dead tree stems) and a dead foliage pool (representing the red needle phase) in RHESSys (Fig.

4). All leaf biomass (including carbon and nitrogen) become part of dead foliage pools. After one

year (Hicke et al. 2012; Edburg et al. 2011) the dead foliage is transferred to litter pools at an

exponential rate with a half-life of two years (Edburg et al. 2012). Similarly, stem carbon and

nitrogen are moved to the snag pool immediately after outbreak. After five years (Edburg et al.

2012), carbon and nitrogen in snags begin to move into the coarse woody debris (CWD) pool at



an exponential decay rate with a half-life of ten years (Edburg et al. 2011). After outbreak, the
coarse root pools that are killed move to the CWD and fine root pools move to litter pools. To
simplify, we assume a uniform mortality level for all evergreen patches across landscape. Due to
the limitation of land cover data, we cannot separate pine and fir in these evergreen patches.
However, this will not affect the interpretation of our results because we analyze them based on
mortality level and evergreen vegetation coverage rather than different species.
In the integrated model, the reduction of leaf carbon and nitrogen after beetle outbreak can
directly decrease LAI and canopy height, which consequently affects energy (i.e., longwave
radiation and the interception of shortwave radiation) and hydrologic (i.e., transpiration and
canopy interception) fluxes. We calculate two types of LAI: **Live LAI** (i.e., only live leaf is
included), and **Total LAI** (i.e., both live and dead leaves are included). The calculation of plant
transpiration is based on Live LAI, while the calculation of other canopy properties, including
interception and canopy evaporation, is based on Total LAI. The calculation of canopy height
includes the living stem and the snag pool.
3.3 Input data
We used the US Geologic Survey (USGS) National Elevation Dataset (NED) at 10 m resolution
to calculate the topographic properties of Trail Creek, including elevation, slope, aspect, basin
boundaries, sub-basins, and patches. Using NED, we delineated 16705 100-m resolution patches
within 72 sub-basins. We used the National Land Cover Database (NLCD) to identify five
vegetation and land cover types, i.e., evergreen, grass/herbaceous, shrub, deciduous, and urban
(Homer et al. 2015). We determined soil properties for each patch using the POLARIS database
(probabilistic remapping of SSURGO; Chaney et al. 2016). Parameters for soil and vegetation



were based on previous research and literature (White et al. 2000; Law et al. 2003; Ackerly
2004; Berner and Law 2016; Hanan et al. 2016).
Climate inputs for this study, including maximum and minimum temperatures, precipitation,
relative humidity, radiation, and wind speed, were acquired from gridMET for years from 1980
to 2018. GridMET provides daily high-resolution (1/24 degree or ~4 km) gridded meteorological
data (Abatzoglou 2013). It is a blended climate dataset that combines the temporal attributes of
gauge-based precipitation data from NLDAS-2 (Mitchell et al. 2004) with the spatial attributes of
gridded climate data from PRISM (Daly et al. 1994).
3.4 Simulation experiments
To quantify how water yield responds to beetle-caused mortality, we designed the following
simulation experiment. We prescribed a beetle outbreak in September 1989, the mortality level
(%) is applied to all evergreen patches for each sub-basin. After beetle outbreak, red needles stay
on the trees for one year before they start to fall (transferred to the litter pool) at an exponential
rate with a half-life of two years. The snag pools stay in the standing trees for five years and then
start to fall and are added to the CWD pool which decays at an exponential rate with a half-life
of ten years.
To address Q1 (i.e., the role of interannual variability), we compared water yield responses
during a dry water year, 1994 (i.e., five years after beetle outbreak with precipitation 611 mm),
to responses during a wet year, 1995 (i.e., six years after beetle outbreak with precipitation 1394
mm). This enabled us to estimate the role of interannual climate variability in driving changes in
water yield following beetle attack. The dry year are selected based on years that have
precipitation below the 15[th] percentile across 38 years of annual precipitation data (from 1979 to
2017) (Searcy 1959, see Fig. S1). During these early period after beetle outbreak (e.g., 1994 and
1995) the forest is experiencing large changes in vegetation canopy cover, plant transpiration,
and soil moisture. We chose these two successive years because they have almost similar canopy
and vegetation status in terms of fallen dead foliage and residual vegetation regrowth, which
makes this comparison reasonable. However, it is possible that antecedent climate conditions
may affect the following year's response. For example, soil moisture can be depleted during a
drought year, affecting initial conditions the following year. Moreover, under drought conditions,
less reactive nitrogen is taken up by the plants or leaching is reduced, so more nitrogen will be
left for the following year. Therefore, the difference in water yield responses between 1994 and
1995 might be affected by not only climate variations but also initial conditions in the hydrology
and the biogeochemistry. To consider the time lag effect (antecedent conditions affecting the
current year's response), we also analyzed other dry and wet years.
To address Q2 (i.e., the role of vegetation mortality), we prescribe a range of infestation-caused
mortality levels (i.e., from 10% to 60% by a step of 10% in terms of carbon, uniformly applied to
all evergreen patches for each sub-basins ) and a control run (no mortality) to quantify the
response of forests in water yield to vegetation mortality level (for each sub-basin **vegetation**
**mortality** is **evergreen mortality** multiplied by evergreen coverage of that basin). The
differences in water yield between each mortality level and the control run represent the effects
of beetle kill: a positive value means that mortality increased water yield, and vice versa.
We quantified the water budget for each sub-basin to examine which hydrological process
contribute to the water yield responses: water yield (Q), precipitation (P), canopy evaporation
($E_{canopy}$, canopy evaporation and snow sublimation), transpiration (T), ground evaporation
($E_{ground}$, includes bare soil evaporation, pond evaporation, and litter evaporation), snow
sublimation (Sublim, ground), soil storage change ( $dS_{soil}/dt$ ), litter storage change





$({}^{dS_{litter}}/_{dt})$, snowpack storage change $({}^{dS_{snowpack}}/_{dt})$  and canopy storage change
$({}^{dS_{canopy}}/_{dt})$.
The storage components include soil, litter, and canopy. According to Eq. (1), if the storage
increases, water yield decreases.
$$Q = P - E_{canopy} - E_{ground} - Sublim -$$
$$T - {}^{d(S_{soil} + S_{litter} + S_{canopy} + S_{snowpack})}/_{dt} \qquad (1)$$
Calculating water balance differences between different mortality scenarios and control scenario
results in Eq. (2):
$$\Delta Q = \Delta E_{canopy} + \Delta E_{ground} + \Delta Sublim + \Delta T +$$
$$\Delta({}^{d(S_{soil} + S_{litter} + S_{canopy} + S_{snowpack})}/_{dt}) \qquad (2)$$
To address Q3 (i.e., the role of long-term aridity), we calculated the long-term aridity index
(PET/P, Fig. 3) across the basin and analyzed the relationship between long-term aridity index
and hydrologic response. As mentioned earlier, the long-term aridity index is defined as the ratio
of mean annual potential ET (PET) to annual precipitation (P), averaged over 38 years (water
year 1980-2018) of historical meteorological data. Based on the long-term aridity index, we
classified our sub-basins into three types ( McVicar et al. 2012, Table 1).





**4 Results**

4.1 Simulated vegetation response to beetle outbreak at basin-scale

4.1.1 Vegetation response to beetle outbreaks

Figure 5 shows the basin-scale vegetation response after beetle outbreak in 1989. Live LAI
dropped immediately after beetle outbreak, then gradually recovered to pre-outbreak levels
during following years (Fig. 5a). Total LAI (i.e., including dead foliage) showed a slight increase
during the first ten years after beetle outbreak (1990 – 2000), which is due to the retention of
dead leaves in the canopy and the simultaneous growth of residual (unaffected) overstory and
understory vegetation (Fig. 5b). The dead foliage pool (Fig. 5c) remained in place for one year
and then began to fall to ground (converted to litter) exponentially with a half-life of two years,
and the snag pool (Fig. 5d) remained in place for five years and then began to fall to ground
(converted to CWD) exponentially with a half-life of ten years. These behaviors of the dead
foliage and snag pools are similar to Edburg et al. (2012), which demonstrates that the integrated
model is simulating expected vegetation dynamics following beetle outbreak.

4.1.2 Time series of hydrologic response to beetle outbreak

Figure 6 shows the changes in simulated water fluxes and soil moisture over the basin after
beetle outbreak with various evergreen mortality levels. During the first 15 years after beetle
outbreak, scenarios where the evergreen mortality level was larger than zero had higher basin-
scale water yield than the control scenario (where the evergreen mortality level was zero). This
was especially true during wet years; however, there was no significant increase during dry years
(i.e., 1992, 1994, 2001, and 2004; Fig. 6a). The year-to-year soil storage fluxes responded
strongly in the first two years after beetle outbreak, then stabilized to the pre-outbreak condition
(Fig. 6b). Note that year-to-year soil storage change is not the same as soil water storage. After





beetle outbreak, the soil can hold some portion of water that not being up taken by the plants, but
it was confined by the soil water holding capacity. This phenomenon indicates that the soil has
some resilience to vegetation change.
Beetle outbreaks reduced transpiration during wet years but did not have significant effects in
dry years (Fig. 6c). This is because transpiration in dry years was water-limited and so was much
lower than the potential rate (more water is partitioned to evaporation; Biederman et al. 2014).
Thus, killing more trees had little effect on stand scale transpiration because remaining trees
utilized any water released by the dead trees in dry years. On the other hand, plant transpiration
in wet years was close to the potential rate; therefore, decreases in canopy cover reduced
transpiration. The simulation results did not show any apparent effect on snowmelt after beetle
outbreak.
The evaporation response was opposite in dry and wet years: evaporation increased in dry years,
while it decreased in wet years (Fig. 6d). This phenomenon is caused by tradeoffs and
interactions among multiple processes, as will be explained in more detail in the next section.
4.2 The role of spatial heterogeneity in water yield response
4.2.1 Spatial patterns of hydrologic response along long-term aridity gradient
4.2.1.1 Evaporation
Beetle outbreak had opposite effects on evaporation between a dry year and a wet year (Fig. 7).
In the dry year, most sub-basins experienced higher evaporation for beetle outbreak scenarios
than in the control scenario (Fig. 7a). This was the cumulative consequence of decreased canopy
evaporation and increased ground (soil, litter, pond) evaporation due to decreases in LAI (caused
by mortality). In the dry year, the latter effect (i.e., increased ground evaporation) dominated





407 over the former effect so that overall consequence was increased evaporation. When the

408 vegetation mortality level (calculated as *the percentage of evergreen patches in a sub-basin*

409 *multiplied by the mortality level of evergreen caused by beetles*) was higher than 20%, a few sub-

410 basins in the balanced (more mesic) area showed some decrease, indicating that the effects of

411 decreasing canopy evaporation exceeded the effects of increasing ground evaporation. In the wet

412 year, most of the sub-basins located in the balanced area showed decreases in evaporation, and

413 the decreasing trend showed linear relationship with vegetation mortality level (where canopy

414 evaporation decreases are dominant, Fig. 7b). However, sub-basins located in much drier regions

415 (aridity >3.5) had relatively insignificant responses to vegetation mortality levels and some of

416 them even had slight increases in evaporation (where ground evaporation increases are dominant

417 due to drier long-term climate and less pine coverage resulted in lower canopy mortality).

418 4.2.1.2 Transpiration

419 Beetle outbreak decreased transpiration in both dry and wet years, and with higher mortality

420 levels the decrease became larger (Fig. 8). However, during the dry year, the water-limited area

421 showed less change than the balanced area; some sub-basins even showed slight increases. This

422 increase in the water-limited part of the basin occurred because after beetles kill some overstory

423 evergreen trees, the living trees and understory plants together can exhibit higher transpiration

424 rates in dry years (Tsamir et al. 2019). In the wet year, when most canopies reach potential

425 transpiration rates (less competition for water), beetle outbreaks can reduce transpiration rates by

426 decreasing Live LAI.

427 4.2.1.3 Total ET





Figure 9 depicts the spatial pattern of changes in total ET (i.e., evaporation and transpiration)
after beetle outbreak. In a dry year, the balanced and water-limited areas showed opposite
responses to mortality: the balanced area showed a decrease in ET and the water-limited area
showed a slight increase. In the balanced area, larger ET decreases occurred with higher
mortality levels. However, increases in ET in water-limited regions were less sensitive to
vegetation mortality level, and even for high vegetation mortality levels (>40%), ET still
increased (Fig. 9a). During the wet year, most sub-basins experienced decreasing ET after beetle
outbreak and the magnitude was larger with higher vegetation mortality. The different responses
of ET were driven by different hydrologic responses (transpiration, ground evaporation and
canopy evaporation) competing with each other; this competition was influenced by climate
conditions, mortality level, and spatial heterogeneity in long-term aridity.
4.2.1.4 Water yield
In the dry year (1994), beetle-caused vegetation mortality affected water yield (Fig. 10), but the
responses differed between the balanced and water-limited areas. For the **balanced area**, most
sub-basins showed slight decreases in water yield after beetle outbreak and no significant
differences among low vegetation mortality level (<=40%, Fig. 10a). However, with increased
mortality levels, more sub-basins showed increases in water yield, particularly with vegetation
mortality higher than 40% (Fig. 10a). Moreover, the vegetation mortality threshold that changed
the direction of water yield response was altered by long-term aridity, e.g., it was 40% for aridity
2.0 but 20% for aridity 1.0.  For **the water-limited** area, water yield decreased and was
independent from mortality level (Fig. 10a). In the wet year (1995), the water yield in most sub-
basins increased after beetle outbreak, and the balanced area increased more significantly than
the water-limited area. Furthermore, for the balanced area, higher mortality levels caused larger





increases in water yield which responded more linearly (Fig. 10b). In summary, for a wet year,
increases in water yield occurred for most sub-basins, driven by a decrease in ET. However,
during dry years, the water yield and ET responses were spatially heterogeneous, and the
competing changes in evaporation and transpiration changed the direction and magnitude of ET
and thus water yield response. The competing effect among different hydrologic fluxes for a dry
year is explored in more detail in the next section.
4.2.2 Water budgets to understand decreasing water yield in the dry year
We analyzed the fluxes in greater detail in a dry year (1994) to understand the response of
hydrologic fluxes and resulting water yield. Based on Eq. (2), we identified four hydrological
fluxes that can potentially affect water yield: canopy evaporation (canopy evaporation and
canopy snow sublimation), ground evaporation (bare soil evaporation, ground snow sublimation,
litter evaporation, pond evaporation), plant transpiration, and year-to-year storage change (soil,
canopy, litter, snowpack). These three storage terms (canopy, litter, snowpack) were considered
together with soil storage since their contribution was minor in comparison with other fluxes.
Figure 11 summarized different combinations of these four dominate processes during the dry
year (1994) based on their directions (increase or decrease in water yield) after beetle outbreak.
In total, fourteen combinations of changes in these fluxes (referred to as "response types") were
found. Five of them resulted in an increase in water yield, and the others resulted in a decrease.
Water yield responses caused by the competition of different hydrologic fluxes showed different
patterns across the aridity gradient (Figs. 3&10). For the balanced area (upper part of the basin),
with low evergreen mortality (<=30%), the major response types were D1 and D2, in which the
increase in ground evaporation dominated over the decrease in transpiration and canopy
evaporation (Fig. 11a, b, and c). However, with higher evergreen mortality (>30%), the major





response type became W2, where the increase in ground evaporation did not exceed the decrease
in canopy evaporation and transpiration (Fig. 11e, f, and g). This indicates that, in a dry year,
when more evergreen stands are killed, the increase in ground evaporation reaches a limit while
transpiration and canopy evaporation continue to decrease with decreasing LAI. The increase in
ground evaporation was triggered either by decreased Total LAI and open canopy, which
allowed more solar radiation penetration to the ground for evaporation (Fig. S5c), or less
transpiration from plants, which left more water available to evaporate (Fig. 8a). The decrease in
plant transpiration and canopy evaporation was driven by a lower Live LAI and a lower Total
LAI, respectively (Fig. S5 a&c and Fig. 8a).
The decrease in water yield in the water-limited area (lower part of the basin) was driven by
different hydrologic flux competitions in different mortality levels. When evergreen stand
mortality level was low (<=30%), the response types were D5 and D7, in which the increase in
ground and canopy evaporation dominated over the decrease of transpiration (Fig. 11a, b, and c).
However, with high evergreen stand mortality (>30%), the response types became D1 and D2
(Fig. 11e, f, and g), in which the canopy evaporation changed from an increase to a decrease that
was driven by a decrease in Total LAI (Fig. S5c). When mortality was low, the increases in
growth from residual plants and understory outstripped the litter fall of dead foliage; thus, Total
LAI increased, and vice versa when mortality was high.
**5 Discussion**
5.1 Role of interannual climate variability
During the first 15 years after beetle attack, various hydrologic processes opposed and/or
reinforced one another to either increase or decrease water yield: a decrease in Live LAI can
reduce transpiration, while a decrease in Total LAI can enhance ground evaporation but diminish



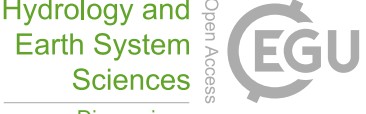

canopy evaporation (Montesi et al. 2004; Tsamir et al. 2019). Interannual climate variability
played an important role in determining which of these competing effects dominate and,
therefore, drove the direction of water yield response to beetle outbreak (Winkler et al. 2014;
Goeking and Tarboton 2020). Our results show that mainly decreases in water yield occurred in
dry years, while increases occurred in wet years. During a wet year, plant ET can reach its
potential so that any reductions in actual plant ET will dominate over any increases in ground
evaporation, resulting in a net increase in water yield. During a dry year, the relative dominance
of these competing effects had greater spatial heterogeneity because the water stress status of the
plants varied across the basin (as explained in Sect 4.2.2; Fig. 11).
However, the responses we observed in the dry year (1994) and in the wet year (1995) were also
affected by the previous year's climate (mainly precipitation) and its effects on hydrologic and
biogeochemical processes, which set the initial conditions for the dry and wet year (e.g., soil
moisture, nitrogen availability, etc.). Therefore, we also analyzed other water years during the
first ten years after beetle outbreak to examine whether our findings for dry and wet years follow
a general pattern and to what extent they are influenced by antecedent conditions. Results
indicate that our findings are robust through the study time period. For example, water yield
generally decreased during dry years (1992, 1994, and 2001, see Figs. S1 and S2) and always
increased during wet years (1993 and from 1995 to 2000, see Fig. S1 and S2).
Adams et al. (2012) provide a threshold of precipitation under which water yield increases after
disturbances: at least 500 mm/year (Goeking and Tarboton 2020). The average annual
precipitation over this study basin is 600-900 mm in dry years, and higher than 900 mm in wet
years. Recent field work observation also find annual climate variability can affect the magnitude
of evapotranspiration fluxes that change the water yield direction (Biederman et al. 2014). Our





results corroborate these earlier studies by revealing that there are precipitation thresholds above
which tree removal increases water yield (Figs. 10, S1 and S2).

## 5.2 Role of vegetation mortality

Vegetation mortality is another important factor that influences water yield response. We found
that during the wet year, beetle outbreak increased water yield across the basin and the
magnitude of these increases grew linearly with the level of vegetation mortality (Fig. 10b). In
the dry year, however, the response of water yield to the level of vegetation mortality was more
complicated because mortality influenced not only the magnitude of change but also the
direction (Fig. 10a). These opposing results (due to mortality level) mainly occurred in the
"balanced" northern part of the basin, where the competing effects of mortality (i.e., increases in
ground evaporation versus decreases in transpiration) are more balanced (Fig. 11). The level of
vegetation mortality played a less significant role in changing water yield in the southern "water-
limited" area. Vegetation mortality level determined the magnitudes of Live LAI, Total LAI,
transpiration, canopy evaporation, and ground evaporation in such a way that it governed the
direction of change in both ET and water yield. Thus, when vegetation mortality level was higher
than 40%, its effect of decreasing transpiration became the dominant process and its effect of
increasing soil evaporation became minor (Fig. 11 f&g; Guardiola-Claramonte et al. 2011).
Besides the precipitation threshold of at least 500 mm/year, Adams et al. (2012) also estimate
that when at least 20% of vegetation cover is removed, water yield can increase. According to
previous analysis (Sect 4.1), for a dry year, water yield increases when more than 40% of
vegetation is removed (Fig. 10a). Our model simulations indicate similar mortality thresholds
exist for driving water yield increases during the dry year, however, we did not find evidence
that such a threshold exists during wet years. These differences between dry and wet years





suggest that the effects of mortality on water yield depend on climate variability. Other studies
corroborate this finding by demonstrating that the relationship between mortality level and water
yield response is complicated and nonlinear (Moore and Wondzell 2005).
5.3 Role of long-term aridity index (PET/P)
Long-term aridity indices can be used to predict where water yield will decrease after
disturbance. We found that water yield always increased in a wet year, irrespective of the
climatic aridity index (Fig. 10a). For dry years, long-term aridity index became important in
driving the direction of water yield responses to beetle outbreak. In areas that are less water-
limited (balanced areas), the direction of water-yield responses to beetle outbreak in a dry year
was mixed and depended on mortality level. For water-limited areas, in a dry year, water yield
showed a more consistent decrease trend, and it was also less affected by mortality level. These
results agree with previous studies finding that water yield decreases largely happen in semiarid
areas (Guardiola-Claramonte et al. 2011; Biederman et al. 2014).
The decrease in water yield for water-limited area can be driven by increases in canopy
evaporation or transpiration, which were different in the hydrologically-balanced area (driven by
increase of ground evaporation). There, the increase in canopy evaporation was due to an
increase in total LAI which is a combined effect of delayed decay of dead foliage and fast
growth of residual and understory plants (Fig. 11d type D5,  D7, D8 & D9; Fig. S5). The
surviving and understory plants in the water-limited area can also have higher transpiration rates
after mortality (Fig. 11d type D6 and Fig. 8). Similarly, in field studies, Tsamir et al. (2019)
found an increase in photosynthesis and transpiration after thinning in a semi-arid forest. These
findings illustrate that in addition to top-down climate variability, the long-term aridity index




(which also varies with bottom-up drivers such as vegetation and local topography) can be
another useful indicator of how water yield will respond to disturbances.
5.4 Uncertainties
While our findings revealed how topoclimatic gradients influenced water yield responses to
beetle infestation, some uncertainties remain. For one, we used uniform mortality levels for all
patches across the watershed rather than location and vegetation-specific mortality levels.
However, in reality beetles usually attack older trees first (Edburg et al. 2011). Thus,
incorporating a more mechanistic understanding of beetle attack patterns with our beetle effects
model could enable us to simulate more realistic outbreak scenarios moving forward. Another
source of uncertainty came from the model treatment of litter pools. In the current
implementation, we ignored the effects of litter on ground albedo and snowmelt (Lundquist et al.
2013), which could have an effect on rates of AET and PET and therefore our calculated long-
term aridity index. Also, because we focused on water yield responses during the first 15 years
after beetle outbreak, we may have missed some of the long-term effects (e.g., after the
ecosystem has begun to recover) on forest hydrology. Future research should integrate the short-
term and long-term effects and interactions among beetle outbreak, vegetation dynamics, and
hydrology. Since Trail Creek is either "balanced" or "water-limited" in terms of aridity, other
"energy-limited" regions could also be investigated.
**6 Conclusion**
We tested a coupled ecohydrologic and beetle effects model in a semi-arid basin in southern
Idaho to examine how watershed hydrology responds to beetle outbreak and how interannual
climatic variability, vegetation mortality, and long-term aridity influence these responses.
Simulation results indicate that each factor can play a discrete role in driving hydrological





processes (e.g., the direction and magnitude of changes in plant transpiration, canopy and soil
evaporation, soil and litter moisture, snow sublimation, etc.). These combined effects determine
the overall water budget and water yield of the basin. While interannual climate variability is the
key factor driving the direction of change in water yield, vegetation mortality levels and long-
term aridity modify water yield responses.
In dry years, the water yield of most sub-basins slightly decreased after beetle outbreak when
vegetation mortality level was lower than 40%; while during wet years in most sub-basins it
increased. Our results show that long-term aridity index is a reliable indicator of the water yield
decreases that occur during dry years due to the fact that there is a consistent decrease in water
yield in the most water-limited portion of the basin. Generally, the effects of vegetation mortality
on water yield during dry years is less uniform and depends on local, long-term aridity
conditions. During wet years, on the other hand, mortality typically causes increases in water
yield. This illustrates that together interannual climate variability and mortality can have a
stronger effect on the direction of water yield response in water-limited regions than interannual
climate variability alone. Future studies to predict water yield response to disturbance should
consider the interactions of these factors and capture the fluctuations of competing water fluxes
and storage change that control overall water budget and water yield.
Using our novel RHESSys-beetle effects modeling framework, we demonstrate that the direction
of hydrologic response is a function of multiple factors (e.g., interannual climate variability,
vegetation mortality level, and long-term aridity) and that these results do not necessarily *conflict*
with each other but are representative of different conditions. The mechanisms behind these
changes compete with each other resulting in a water yield increases or decreases (Fig. 1).
Contradictory findings in previous studies may result from differing mortality levels (disturbance



severity), or differences in aridity, and consequently, the emergent drivers that dominate water
yield responses differ. Disentangling these drivers is difficult or impossible using a purely
empirical approach where it can be challenging or cost-prohibitive to experiment under a broad
range of controlled conditions. Distributed process-based models on the other hand, provide a
useful tool for examining these dynamics.
Findings from this study can assist water supply stakeholders in risk management in beetle
outbreak locations. For example, during wet years, more attention might be focused on
"balanced" areas, i.e., wet regions, for flooding and erosion risks after beetle outbreaks since
these regions may experience large increase in runoff due to decreases in plant transpiration and
increases in soil moisture. During the dry years, attention might need to shift to "water-limited"
areas for managing wildfire risk since these regions will experience elevated ET and lower soil
and litter moisture. Because multiple factors interact to influence hydrological processes after
beetle outbreak, water and forests management must respond to spatial and temporal variations
in climate, aridity, and vegetation mortality levels.
**Code and data availability**
The coupled RHESSys model code is available online at:
https://github.com/renjianning/RHESSys/tree/historical_fire
The data used in this study are available at:
https://osf.io/tsu9z/?view_only=72bfa7b376ad40c59278312f49b03a69
**Author contributions**
JR, JA and JAH conceived of study. JR designed study with support from JA, JAH and EH. JR
and EH developed RHESSys code for coupling beetle effect model and parallelizing model runs



with help from JA, JAH, NT, ML, CK, and JTA. JR performed model simulations and developed
figures with help from all authors. ML and JTA generated downscaled meteorological data. JR
wrote manuscript with input from all authors.
**Competing interests**
The authors declare that they have no conflict of interest.
**Acknowledgments**
This project is supported by National Science Foundation of United States under award number
DMS-1520873.



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

9326/11/7/074010.
Smith, Frederick W., D. Arthur Sampson, and James N. Long. 1991. "Comparison of Leaf Area
Index Estimates from Tree Allometrics and Measured Light Interception." *Forest Science*
37 (6): 1682–88. https://doi.org/10.1093/forestscience/37.6.1682.
Smith, Rex Onis. 1960. "Geohydrologic Evaluation of Streamflow Records in the Big Wood
River Basin, Idaho." USGS Numbered Series 1479. Water Supply Paper. U.S. Govt.
Print. Off.,. http://pubs.er.usgs.gov/publication/wsp1479.
Son, Kyongho, and Christina Tague. 2019. "Hydrologic Responses to Climate Warming for a
Snow-Dominated Watershed and a Transient Snow Watershed in the California Sierra."
*Ecohydrology* 12 (1): e2053. https://doi.org/10.1002/eco.2053.
Sun, Ning, Mark Wigmosta, Tian Zhou, Jessica Lundquist, Susan Dickerson-Lange, and
Nicoleta Cristea. 2018. "Evaluating the Functionality and Streamflow Impacts of
Explicitly Modelling Forest–Snow Interactions and Canopy Gaps in a Distributed
Hydrologic Model." *Hydrological Processes* 32 (13): 2128–40.
https://doi.org/10.1002/hyp.13150.
Tague, C. L., and L. E. Band. 2004. "RHESSys: Regional Hydro-Ecologic Simulation System—
An Object-Oriented Approach to Spatially Distributed Modeling of Carbon, Water, and
Nutrient Cycling." *Earth Interactions* 8 (19): 1–42. https://doi.org/10.1175/1087-
3562(2004)8<1:RRHSSO>2.0.CO;2.
Tague, Christina L., Max Moritz, and Erin Hanan. 2019. "The Changing Water Cycle: The Eco-
Hydrologic Impacts of Forest Density Reduction in Mediterranean (Seasonally Dry)
Regions." *Wiley Interdisciplinary Reviews: Water* 0 (0): e1350.
https://doi.org/10.1002/wat2.1350.
Tsamir, Mor, Sagi Gottlieb, Yakir Preisler, Eyal Rotenberg, Fyodor Tatarinov, Dan Yakir,
Christina Tague, and Tamir Klein. 2019. "Stand Density Effects on Carbon and Water
Fluxes in a Semi-Arid Forest, from Leaf to Stand-Scale." *Forest Ecology and*
*Management* 453 (December): 117573. https://doi.org/10.1016/j.foreco.2019.117573.
White, Joseph D., and Steven W. Running. 1994. "Testing Scale Dependent Assumptions in
Regional Ecosystem Simulations." *Journal of Vegetation Science* 5 (5): 687–702.
https://doi.org/10.2307/3235883.
White, Michael A., Peter E. Thornton, Steven W. Running, and Ramakrishna R. Nemani. 2000.
"Parameterization and Sensitivity Analysis of the BIOME–BGC Terrestrial Ecosystem
Model: Net Primary Production Controls." *Earth Interactions* 4 (3): 1–85.
https://doi.org/10.1175/1087-3562(2000)004<0003:PASAOT>2.0.CO;2.
Wine, Michael L, Daniel Cadol, and Oleg Makhnin. 2018. "In Ecoregions across Western USA
Streamflow Increases during Post-Wildfire Recovery." *Environmental Research Letters*
13 (1): 014010. https://doi.org/10.1088/1748-9326/aa9c5a.
Winkler, Rita, Sarah Boon, Barbara Zimonick, and Dave Spittlehouse. 2014. "Snow
Accumulation and Ablation Response to Changes in Forest Structure and Snow Surface
Albedo after Attack by Mountain Pine Beetle." *Hydrological Processes* 28 (2): 197–209.
https://doi.org/10.1002/hyp.9574.
Zhang, Ke, John S. Kimball, Qiaozhen Mu, Lucas A. Jones, Scott J. Goetz, and Steven W.
Running. 2009. "Satellite Based Analysis of Northern ET Trends and Associated
Changes in the Regional Water Balance from 1983 to 2005." *Journal of Hydrology* 379
(1): 92–110. https://doi.org/10.1016/j.jhydrol.2009.09.047.





Zhao, Maosheng, Steven W. Running, and Ramakrishna R. Nemani. 2006. "Sensitivity of
Moderate Resolution Imaging Spectroradiometer (MODIS) Terrestrial Primary
Production to the Accuracy of Meteorological Reanalyses." *Journal of Geophysical
Research: Biogeosciences* 111 (G1). https://doi.org/10.1029/2004JG000004.



*Table 1. Classification of aridity index.*

| Aridity Index (i.e. PET/P) | Type |
|---|---|
| > 2 | Water - limited |
| 0.8 - 2 | Balanced |
| < 0.8 | Energy - limited |






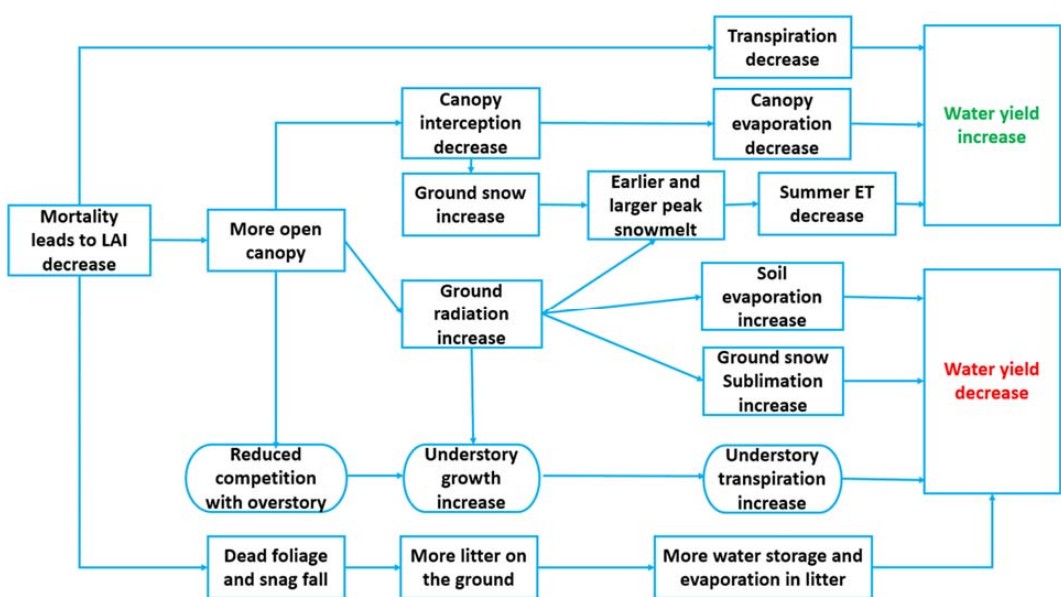


*Figure 1. Mechanism of water yield responses to beetle-caused mortality during the red and*
*gray phases (0 – 10 years after beetle outbreak), semicircle boxes represent understory*
*responses and square boxes represent overstory responses.*

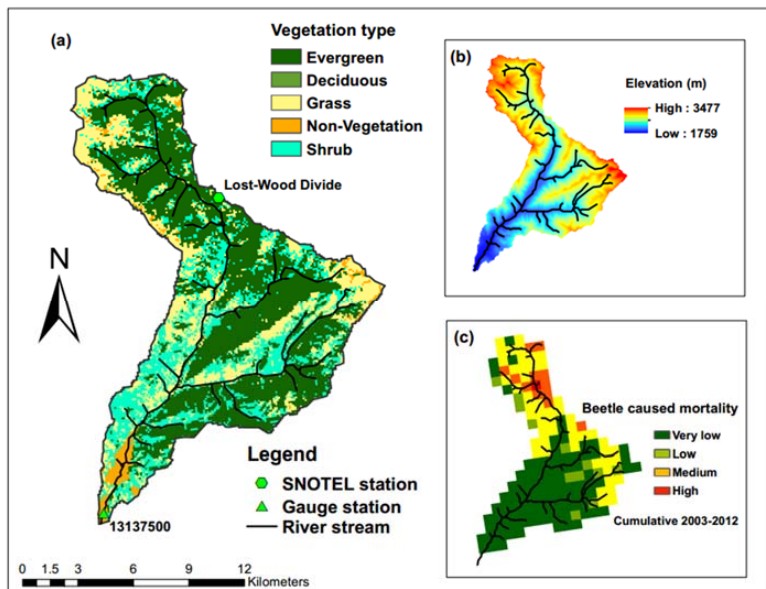


*Figure 2. Land cover, elevation, and tree mortality for Trail Creek. (a) is the land cover map
with the main vegetation type, (b) is the elevation gradient, and (c) is the severity of beetle
caused tree mortality (during the period 2003-2012 Meddens et al. (2012)). Note that, for our
modeling experiments, we prescribe beetle outbreak uniformly across evergreen patches instead
of using historical beetle outbreak data.*



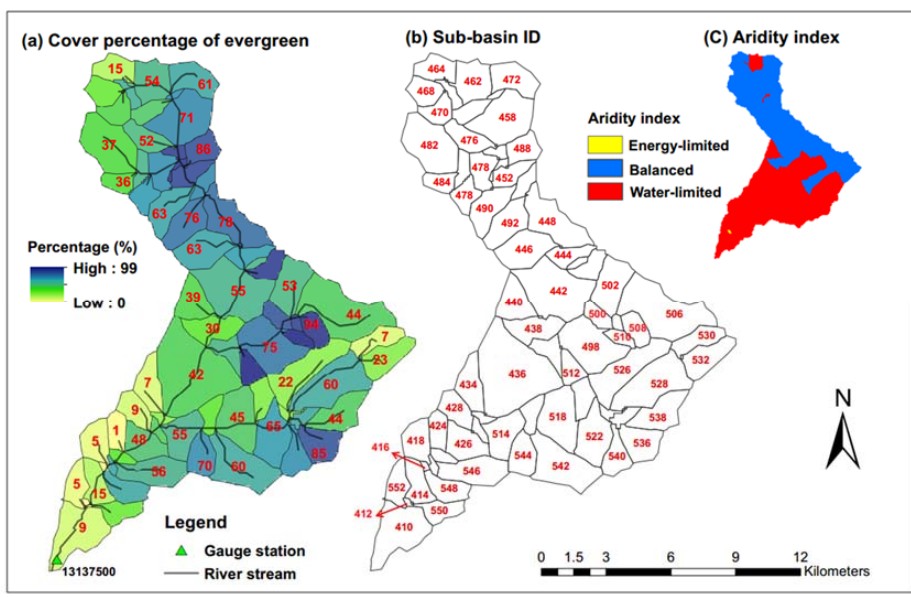


Figure 3. Trail creek evergreen forest cover percentage for each sub-basin, sub-basin ID, and
long-term aridity index. Aridity index is defined as annual mean potential evapotranspiration
(PET) / precipitation (P) from 38 years of data (see Sect 3.4), PET/P > 2 is water-limited, PET/P
< 0.8 is energy-limited, PET/P between 0.8 and 2 is balanced. Recall that only evergreen forest
trees are attacked during beetle outbreaks.






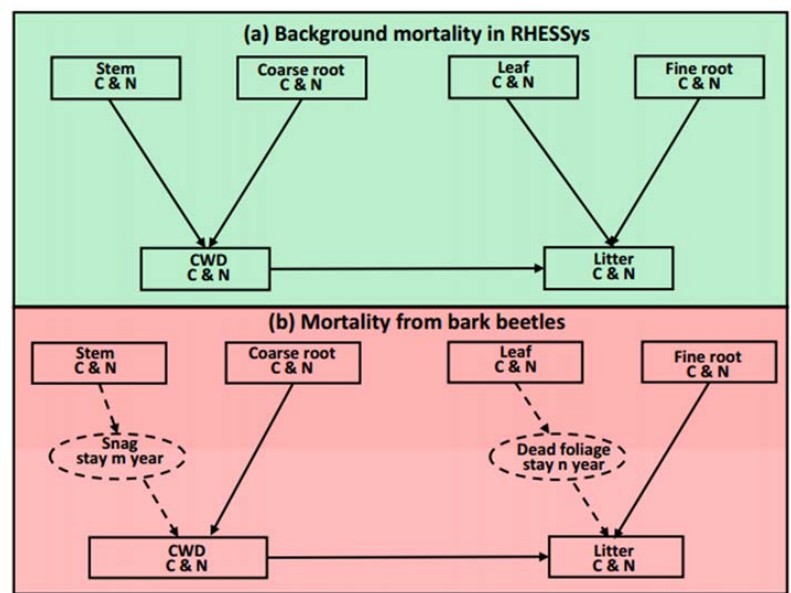


*Figure 4. Conceptual framework of the beetle effect model.*
*(a) Normal background mortality routine in RHESSys before beetle outbreak. (b) Mortality from*
*bark beetles. We add snag (standing dead trees) and dead foliage (needles still on dead trees)*
*pools, shown in the dashed circle. After a beetle outbreak, carbon (C) and Nitrogen (N) move*
*from stems to snag pools (black dashed arrow). After staying in the snag pool for m years, C and*
*N move from snag to coarse wood debris pools (CWD) with an exponential decay rate to*
*represent the snag fall (gray dashed arrow). It is a similar process for leaf C and N, which move*
*from leaf to dead foliage to litter pools (black dotted arrow). Furthermore, C and N in the CWD*
*and fine root pools move to the litter pool immediately after outbreak (solid black and gray*
*arrows). Figure modified from Edburg et al. (2012).*


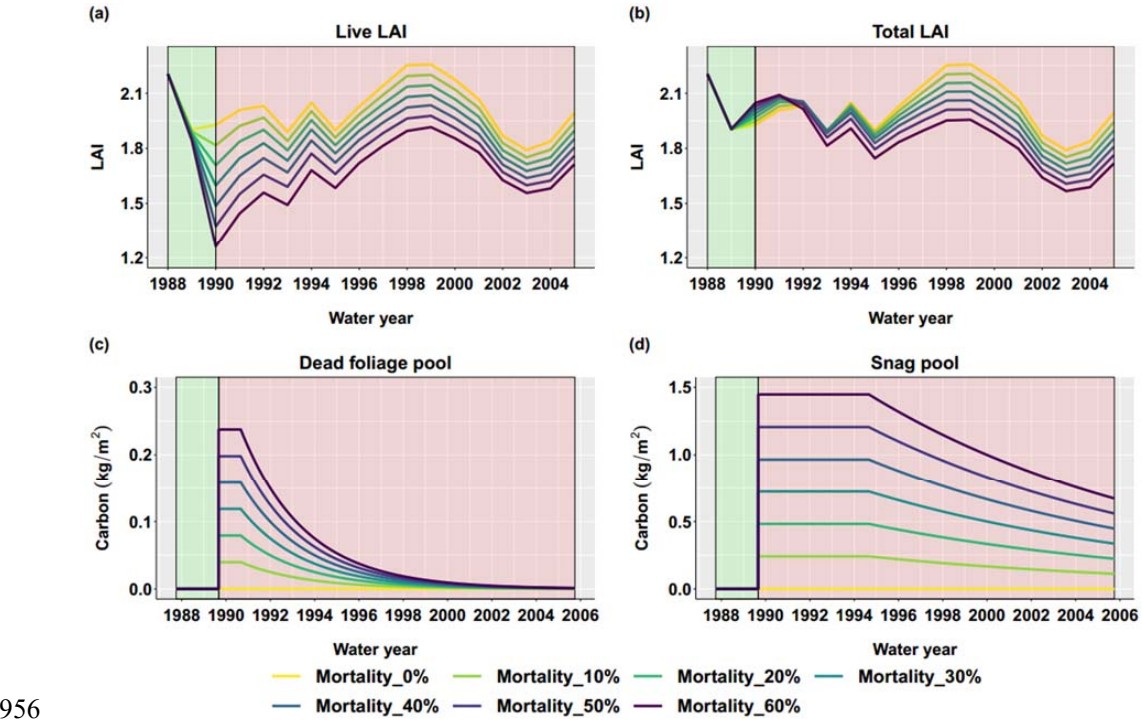


*Figure 5. Basin-scale vegetation responses after beetle outbreak for different evergreen*
*mortality level. (a) Annual live leaf area index (Live LAI), (b) Annual total LAI (LAI calculated*
*including dead foliage pool), and (c) Daily dead foliage pool, and (d) Daily snag pool after outbreak.*
*The green background color is the period before beetle outbreak, and the red background color*
*is after the beetle outbreak.*



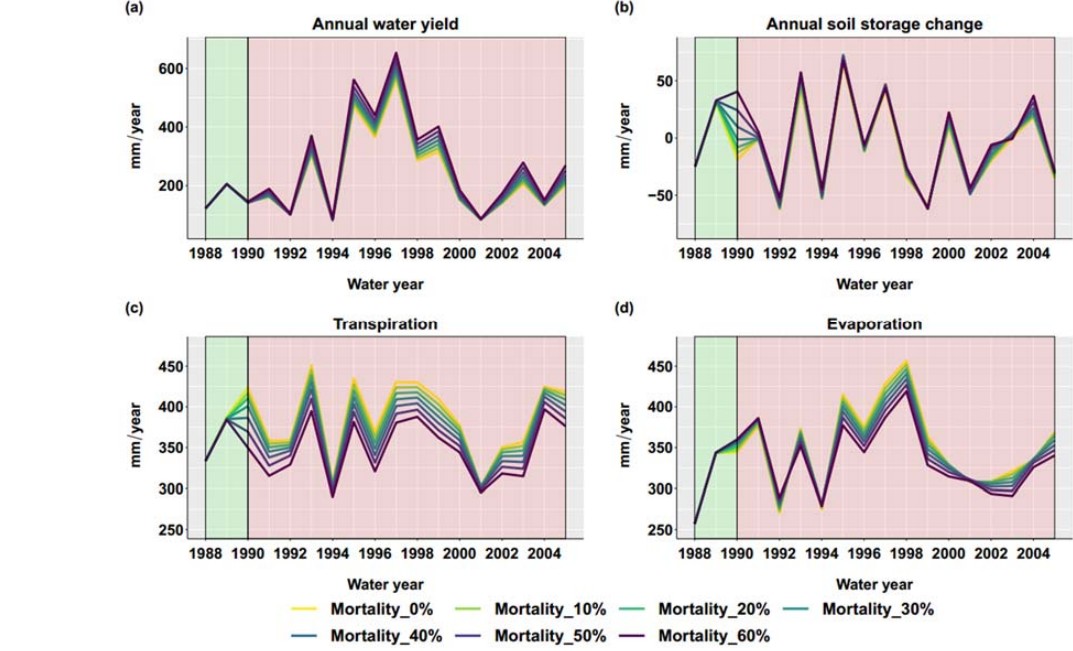


*Figure 6. Basin-scale annual sum of hydrologic fluxes responses after beetle outbreak (1989) for different evergreen mortality levels. (a) Annual water yield calculated as annual sum of basin streamflow, and (b) annual soil water storage change calculated as water year soil water storage at the end of water year minus soil water storage at the beginning of water year. (c) Transpiration is the annual sum of transpiration for both overstory and understory. (d) Evaporation is calculated as the annual sum of canopy evaporation, ground evaporation, and snow sublimation.*

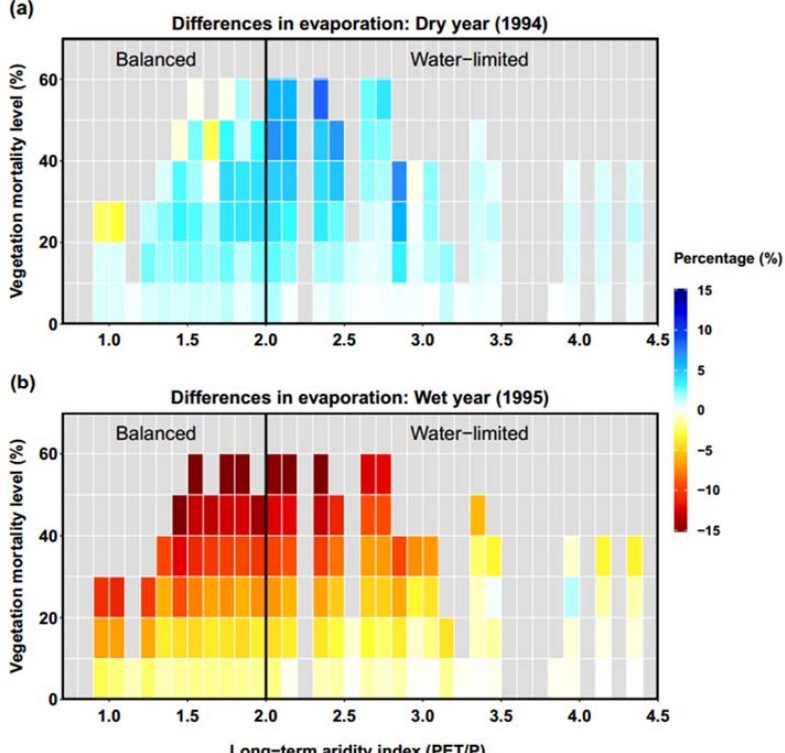

972

*Figure 7. Relationship among long-term aridity, vegetation mortality level, and differences in evaporation for a dry year (1994, a) and wet year (1995, b). Differences are calculated as the normalized differences (%) of evaporation between each evergreen mortality scenario and the control run for no beetle outbreak. Vegetation mortality for each sub-basin is calculated as the percentage of evergreen patches multiplied by the mortality level of evergreen caused by beetles. Long-term aridity is defined as temporally averaged (38 years) potential evapotranspiration relative to precipitation.*

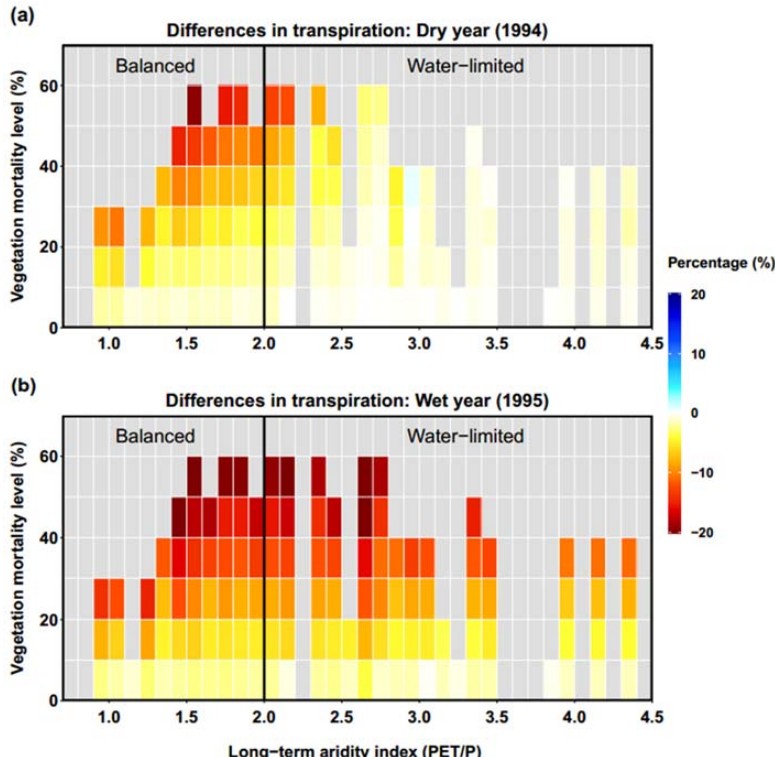


Figure 8. Relationship among long-term aridity, vegetation mortality, and differences in
transpiration for a dry year (1994, a) and wet year (1995, b).

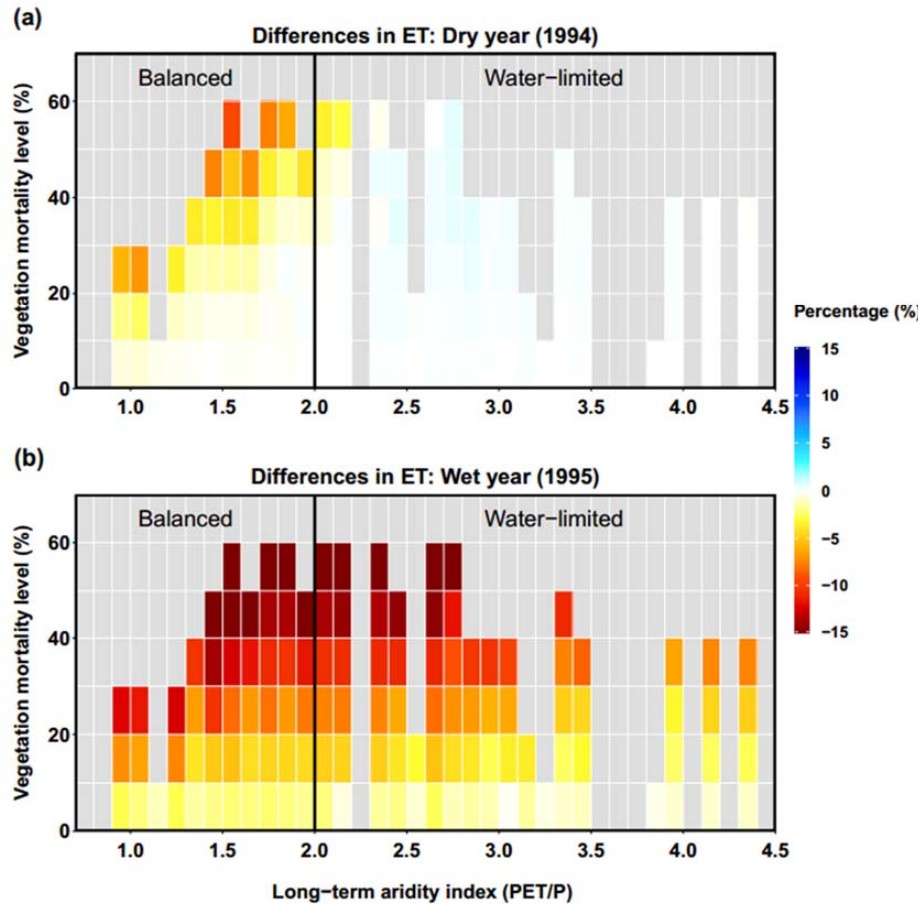


*Figure 9. Relationship among long-term aridity, vegetation mortality level and differences in ET*
*for a dry year (1994, a) and a wet year (1995, b).*




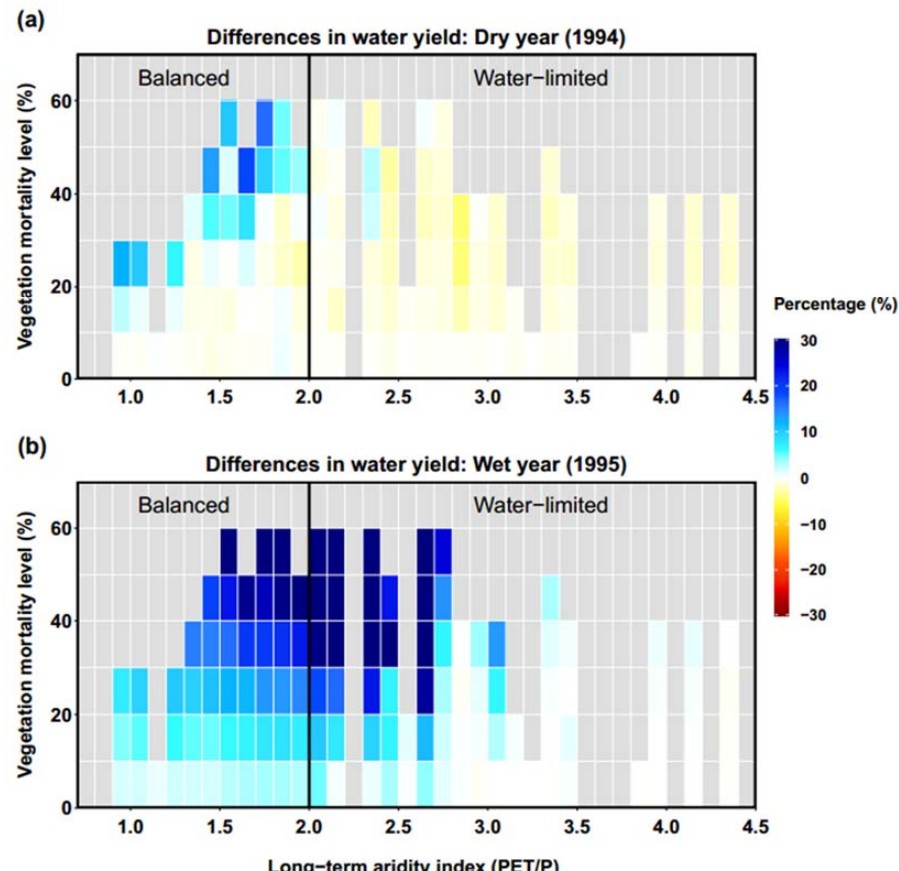


Figure 10. Relationship among long-term aridity, vegetation mortality level and Differences in
water yield for a dry year (1994, a) and wet year (1995, b).




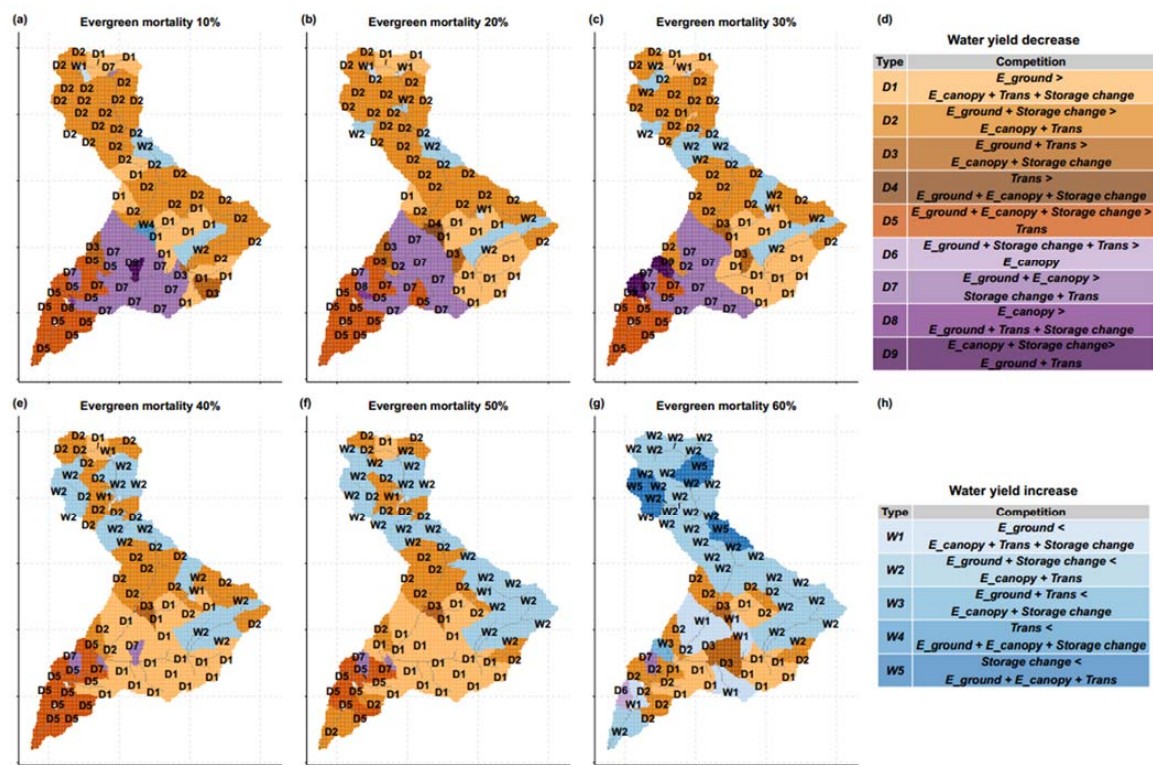


*Figure 11. Water yield response types after beetle outbreak for different evergreen mortality scenarios compared with control scenario. D1 to D9 are water yield decrease types and W1 to W5 are water yield increase types. In panel D and H, the left side of each type are increasing fluxes that cause water yield decreases and the right side are decreasing fluxes that cause water yield increase. If the left side is larger than the right side, water yield increases, and vice versa. (Note: this mortality is evergreen mortality, which is different from vegetation mortality.)*

1000