# Peer review of "How does water yield respond to mountain pine beetle infestation in a semiarid forest?"

_Hydrology and Earth System Sciences, 2020_

## Author Response (AR2)

**Author's Response to Reviewer One**

We thank the reviewer for the helpful comments on the manuscript. We really appreciated the positive feedback. Below is our response to these comments in detail (red text) and text that we will change in the manuscript is in blue text.

**Suggested Improvements**

(1) At the top of page 3 the authors provide a graphical abstract, which is quite helpful to the paper. But I wonder if it is possible to do something similar regarding the physical processes or (water) pathways that are emphasized or de-emphasized or changed from one (temporal + spatial + disturbance) regime to another? I hope this suggestion is clear. As I understand this paper, the authors are describing a systems approach (or model) addressing tree mortality in the Western US. But it seems to me that the paper is largely descriptive of some of the environmental conditions and how they lead to different outcomes for a forest. I think it would be more insightful to discuss the ways in which pathways, by which water moves through the forest ecosystem, change in response to changes in the drivers.

This is a good suggestion. Instead of making a new figure, we have improved the previous graphical abstract by adding a water pathway column.

[Figure]

We updated the abstract to match the graphic abstract: (from line 36 to line 54, the highlighted parts below show these changes)

Mountain pine beetle (MPB) outbreaks in the western United States result in widespread tree mortality, transforming forest structure within watersheds. While there is evidence that these changes can alter the timing and quantity of streamflow, there is substantial variation in both the

magnitude and direction of responses and the climatic and environmental mechanisms driving this variation are not well understood. Herein, we coupled an eco-hydrologic model (RHESSys) with a beetle effects model and applied it to a semiarid watershed, Trail Creek, in the Bigwood River basin in central Idaho to evaluate how varying degrees of beetle-caused tree mortality influence water yield. Simulation results show that water yield during the first 15 years after beetle outbreak is controlled by interactions among interannual climate variability, the extent of vegetation mortality, and long-term aridity. During wet years, water yield after beetle outbreak increases with greater tree mortality; this is driven by mortality-caused decreases in evapotranspiration. During dry years, water yield decreases at low to medium mortality but increases at high mortality. The mortality threshold for the direction of change is location-specific. The change in water yield also varies spatially along aridity gradients during dry years. In relatively wetter areas of the Trail Creek basin, post-outbreak water yield decreases at low mortality (driven by an increase in ground evaporation) and increases when vegetation mortality is greater than 40 percent (driven by a decrease in canopy evaporation and transpiration). In more water-limited areas on the other hand, water yield typically decreases after beetle outbreaks, regardless of mortality level (but the driving mechanisms are different). Results suggest that long-term aridity can be a useful indicator for the direction of water yield changes after disturbance.

(2)  As a follow on from (1) above, I think it would be helpful to move the last paragraph (lines 616-629, page 29)  to the introduction.  It would help to put the modeling effort into context.  Otherwise I was left to wonder until the last paragraph what the application of the authors' modeling effort might be and who might benefit from reading this paper.

Yes, it makes more sense to explain the broader impacts of this paper in the introduction. We will move the last paragraph (lines 616-629, page 29) to the introduction before the conceptual framework section (after line 117, page 7).

(3)  I have no doubt of the importance of aridity in their findings.  Current expectations are that aridity in the western US will continue to worsen as climate change progresses.  Nonetheless, I am having some difficulty with the term "long-term" aridity.  Yes, they do have a 38-year record.  But at present aridity is dynamic (and has the potential to get much worse in far less than 38 years). I think the authors need to recognize and mention in their work that the past record may not be as useful in trying to project into the future as their findings suggest.  Simply extrapolating from the past 38 years of data may bias their results somewhat, especially if aridity (as represented by the aridity index) is prone to rapid intensification.  The paper would benefit by including further discussion of this issue.

Thank you for this suggestion. It prompted us to clarify the strengths and potential limitations of our historical study. To address this concern, we will add an extra paragraph to the discussion section 5.3 - role of long-term aridity index (after line 566; check details below).

We found the long-term (38-year) aridity index for our study region was a key driver influencing hydrologic responses to beetle outbreaks. While this trend is likely to continue in the future as climate change intensifies aridity in the western US (Livneh and Badger 2020), the classification of water-limited/balanced region based on 38-year aridity index may not be stable. Thus, projecting how responses will change under future aridity scenarios requires further modeling research. We used historical 38-years (1980-2019) data to calculate the aridity index (PET/P). This method can be extended to project future responses to beetle outbreaks by using future climate data from generalized circulation models (GCMs) to drive the process-based, ecohydrologic-beetle effects model. Another consideration, however, is that as aridity continues to increase, vegetation may shift from evergreen to more drought-tolerant shrub or grass species. This would in turn alter beetle outbreak patterns and the corresponding water yield responses (Abatzoglou and Kolden 2011; Bart, Tague, and Moritz 2016). However, this process is not well understood and is not currently represented in our modeling framework. Thus, a key uncertainty in predicting future beetle effects is how vegetation will respond to climate change.

(4) In the Introduction the discussion of sublimation should probably cite Frank et al (2019: Bayesian analyses of seventeen winters of water vapor fluxes show bark beetles reduce sublimation. Water Resources Research 55: doi:10.1029/2018WR023054). The findings of Frank et al. (2019) are germane and relevant to the authors' paper. Furthermore, Frank et al. cite other works that the authors should probably cite when discussing the impact that beetles can have on the (canopy-atmosphere-environmental) processes involved in sublimation. Given the importance of snowpacks and sublimation to forest ecosystem water balances I would suggest that the authors provide further discussion of the details of the model's performance regarding sublimation. The model's predictions regarding the change in sublimation (from the snowpack and from the foliage-intercepted snow) before and after the beetles have killed the trees would provide some further insights into how well the model captures these important sublimation-related processes and observations. And although different observers/papers report somewhat different findings, I think comparing the model's predictions of changes in sublimation amounts and stream flow to previous observations would be worthwhile, especially for a systems level model like the one the authors are using.

Thank you for this helpful suggestion. We ran some additional analyses on how snow sublimation responds to beetle outbreak and made the following changes to the text:

1. In the introduction, line 73 (page 5) we will add the following text:

   Snow sublimation is an important process in snow-dominated forest systems. Beetle outbreak can reduce total sublimation, which can in turn increase water yield, especially since canopy sublimation is more sensitive to disturbances than ground snow sublimation (Frank et al. 2019).

2. In the result section 4.1, after line 395 (page 19), we will add an extra paragraph to discuss the sublimation response to beetle outbreak. We also will add Fig S7 to the supplementary material.

Snow sublimation plays an essential role in driving the evaporation responses we observed. In Trail Creek, snow sublimation accounts for around 50% of total evaporation (not shown in the figure), and around 60% is canopy sublimation. Canopy sublimation accounts for an even larger proportion of total sublimation during high snow years (Fig. S7 d and Fig. S1). These results are similar to other Western US forests where 50 to 60% of total sublimation has been found to come from canopy sublimation, which is more sensitive to beetle kill than ground snow sublimation (Molotch et al. 2007; Frank et al. 2019). We also found that during the first three years after beetle outbreak, when dead foliage is still on the canopy, canopy sublimation increases by approximately 6% due to an increase in *Total LAI*, which increases canopy snow interception and subsequent sublimation (Fig. 5). However, when the dead foliage falls to the ground and snags start falling, the canopy sublimation decreases by approximately 10% for the most severe mortality scenario (60% evergreen mortality) compared to no-outbreak scenario. This occurs because canopy *Total LAI* decreases and there is less canopy interception (Fig. 5).

Ground snow sublimation is less sensitive to beetle kill (Fig. S7b). In the first three years after beetle kill (at 60% mortality), ground snow sublimation increases by approximately 7.5% due to an increase of aerodynamic conductance caused by higher understory canopy height. However, from 1993 to 2002, there is no obvious changes in ground snow sublimation after beetle outbreak. When all dead foliage and more than 50% of snags fall to the ground, ground snow sublimation decreases because of increased snowmelt due to the open canopy (Fig. 5 and Fig 7b). In general, for the 60% mortality scenario, the ground snow sublimation first increases by approximately 5% when dead foliage is still on the trees, then decreases by approximately 6% when the canopy is open.

Other recent studies corroborated out findings, although with some discrepancies. Other process-based snow models have shown that canopy sublimation can decrease by 7% and total sublimation can decrease by 4% following beetle attack (Sexstone et al. 2018). Although we find similar decreases trend in canopy snow sublimation after dead foliage falls to the ground, we also find that canopy sublimation increases following beetle outbreak due to an increase in *Total LAI*. This is corroborated by other studies that have found canopy sublimation increases with more leaf area (Koeniger et al. 2008). Furthermore, snowpack sublimation decreases during the open canopy period (caused by increased snowmelt), which is different from other studies that show that snowpack sublimation increases due to a more open canopy (Biederman et al. 2014; Harpold et al. 2014). These contrasts between our research and previous studies indicate the sophisticated balances between canopy-atmosphere-environment when studying the sublimation response to disturbances (Edburg et al. 2012; Frank et al. 2019), which process-based model can help to disentangle.

[Figure]

*Figure S7. Basin-scale snow sublimation responses after beetle outbreak for different evergreen mortality levels. (a) changes (mortality scenario minus control scenario) in terms of percentage in canopy snow sublimation. (b) changes in snowpack sublimation. (c) Changes in total snow sublimation (canopy snow + ground snowpack). (d) the proportion of canopy snow sublimation in total sublimation.*

(5) Lines 344-354, Pages 16-17 – These equations do not make dimensional sense to me. Either Q, P, E and Sublim are rate variables (i.e., mass/unit time) or the storage terms, ΔS, should not be divide by dt. If they are rate variables, the authors should include the physical units in their definition. If they are total amounts (i.e., mass) then they should say that and correct the storage terms.

They are rate variables at an annual time step. We updated equations (1) and (2) at page 17, and the corresponding unit as follow:

$$Q = P - E_{canopy} - E_{ground} - Sublim -$$

$$T - \frac{d(S_{soil} + S_{litter} + S_{canopy} + S_{snowpack})}{dt} \quad (1)$$

**Q**: Water yield (mm/year)

**P**: Precipitation (mm/year)

**E$_{canopy}$**: Canopy evaporation (including canopy snow sublimation, mm/year)

$E_{ground}$: Ground evaporation includes bare soil evaporation, pond evaporation, and litter evaporation (mm/year)

**T**: Transpiration (mm/year)

**Sublim**:  Ground snow sublimation (mm/year)

$dS_{soil}/dt$ : Change in soil water storage calculated at yearly interval

$dS_{litter}/dt$: Change in litter water storage calculated at yearly interval

$dS_{canopy}/dt$: Change in canopy water storage calculated at yearly interval

$dS_{snowpack}/dt$: change in snowpack water storage calculated at yearly interval

Calculating water balance differences between different mortality scenarios and control scenario results in Eq. (2):

$$\Delta Q = \Delta E_{canopy} + \Delta E_{ground} + \Delta Sublim + \Delta T +$$

$$\Delta(d(S_{soil} + S_{litter} + S_{canopy} + S_{snowpack})/dt) \quad (2)$$

**References:**

Abatzoglou, John T., and Crystal A. Kolden. 2011. "Climate Change in Western US Deserts: Potential for Increased Wildfire and Invasive Annual Grasses." *Rangeland Ecology & Management* 64 (5): 471–78. https://doi.org/10.2111/REM-D-09-00151.1.

Bart, Ryan R., Christina L. Tague, and Max A. Moritz. 2016. "Effect of Tree-to-Shrub Type Conversion in Lower Montane Forests of the Sierra Nevada (USA) on Streamflow." Edited by Julia A. Jones. *PLOS ONE* 11 (8): e0161805. https://doi.org/10.1371/journal.pone.0161805.

Biederman, J. A., A. A. Harpold, D. J. Gochis, B. E. Ewers, D. E. Reed, S. A. Papuga, and P. D. Brooks. 2014. "Increased Evaporation Following Widespread Tree Mortality Limits Streamflow Response." *Water Resources Research* 50 (7): 5395–5409. https://doi.org/10.1002/2013WR014994.

Edburg, Steven L., Jeffrey A. Hicke, Paul D. Brooks, Elise G. Pendall, Brent E. Ewers, Urszula Norton, David Gochis, Ethan D. Gutmann, and Arjan JH Meddens. 2012. "Cascading Impacts of Bark Beetle-Caused Tree Mortality on Coupled Biogeophysical and Biogeochemical Processes." *Frontiers in Ecology and the Environment* 10 (8): 416–24. https://doi.org/10.1890/110173.

Frank, John M., William J. Massman, Brent E. Ewers, and David G. Williams. 2019. "Bayesian Analyses of 17 Winters of Water Vapor Fluxes Show Bark Beetles Reduce Sublimation." *Water Resources Research* 55 (2): 1598–1623. https://doi.org/10.1029/2018WR023054.

Harpold, Adrian A., Joel A. Biederman, Katherine Condon, Manuel Merino, Yoganand Korgaonkar, Tongchao Nan, Lindsey L. Sloat, Morgan Ross, and Paul D. Brooks. 2014. "Changes in Snow Accumulation and Ablation Following the Las Conchas Forest Fire, New Mexico, USA: CHANGES IN SNOW FOLLOWING FIRE." *Ecohydrology* 7 (2): 440–52. https://doi.org/10.1002/eco.1363.

Koeniger, Paul, Jason Hubbart, Timothy Link, and John Marshall. 2008. "Isotopic Variation of Snowcover and Streamflow in Response to Changes in Canopy Structure in a Snow-Dominated Mountain Catchment." *Hydrological Processes* 22 (February): 557–66. https://doi.org/10.1002/hyp.6967.

Livneh, Ben, and Andrew M. Badger. 2020. "Drought Less Predictable under Declining Future Snowpack." *Nature Climate Change* 10 (5): 452–58. https://doi.org/10.1038/s41558-020-0754-8.

Molotch, Noah P., Peter D. Blanken, Mark W. Williams, Andrew A. Turnipseed, Russell K. Monson, and Steven A. Margulis. 2007. "Estimating Sublimation of Intercepted and Sub-Canopy Snow Using Eddy Covariance Systems." *Hydrological Processes* 21 (12): 1567–75. https://doi.org/10.1002/hyp.6719.

Sexstone, Graham A., David W. Clow, Steven R. Fassnacht, Glen E. Liston, Christopher A. Hiemstra, John F. Knowles, and Colin A. Penn. 2018. "Snow Sublimation in Mountain Environments and Its Sensitivity to Forest Disturbance and Climate Warming." *Water Resources Research* 54 (2): 1191–1211. https://doi.org/10.1002/2017WR021172.

**Author's Response to Reviewer Two**

We thank the time and effort of the reviewer. These positive feedbacks and discussion are very helpful and much appreciated. Below are the responses in detail (red font) and text that we will change in the manuscript is in blue text.

**Comments**

The authors contributed a very interesting manuscript that is within the scope of the journal and the scientific quality of the ms is very good. Much of my research is in ecohydrology and one recent ms showed the effect of beetle defoliation in dryland riparian corridors of the SW USA and how water use (ET) on these corridors (13 rivers and streams) changed before and after the introduction of the beetles (see Restoration Ecology 2018); therefore, the authors contribution is of great interest to me and certainly contributes something new to the field of hydrology. There are citations that could be added to the ms. background to further demonstrate changes in riparian corridor ET before and after beetle introductions, although their paper is unique in looking at mountain pine beetle infestation and adding in other types of woodlands may not be needed. I have really learned from their discussion and the long-term aridity index is an excellent contribution to water yield research. These results (key points) show that separating wet years and dry years may provide important knowledge that is useful in other systems. I am curious now to apply similar methods in riparian corridors to see if in fact the response to mortality level remains nonlinear and varies by location and year, as I suspect it would in other beetle-infested land covers. My findings suggest that in canopies that were monotypic with high density and extent had increased water yield could be wiped out entirely but then regrown. The regreening post mountain pine beetle does not exist I presume and therefore this work may not be transferrable to other ecosystems, but I do believe this ms and its findings, especially the drought information, is of great interest to the readership. This conclusion was therefore of most interest: " in a dry year, low to medium MPB-caused vegetation mortality decreases water yield, and high mortality increases water yield; this response to mortality level is nonlinear and varies by location and year."

**Thanks for these valuable suggestions and thoughts. We will add more description in the background section starting at line 145 (page 8).**

In some riparian corridors, the regreening of surviving vegetation and the compensatory response of remaining tissues could diminish the reduction in ET caused by foliage fall, leading to no significant water yield response to beetle-caused mortality (Snyder et al. 2012; Nagler et al. 2018).

**Further discussion on testing our method to riparian corridors:**

Riparian corridors may also influence the extent to which mortality and climate variability affect hydrology. We expect that whether ET increases or decreases depends on the competition between higher transpiration rates of surviving vegetation (plus the ground evaporation increase due to open canopy) and lower canopy evaporation and transpiration caused by less canopy foliage. With lower mortality level, the reduction of transpiration (caused by less LAI) can be

small (especially during dry years), and the increase in transpiration rate of surviving plants could be higher, so it may cause an increase in ET or less significant changes in ET. While at high mortality, the reduction in ET caused by less LAI is dominate. Our model results also show an increase in transpiration during dry years caused by higher transpiration in surviving vegetation. This is consistent with a thinning study in a semi-arid forest, where growth rates are 70% higher and transpiration rate are 10% higher after thinning (Tsamir et al. 2019). However, our study site is a snow-dominated watershed and canopy snow sublimation plays an important role in the hydrological response to mortality, indicating that our findings may not be transferable to the riparian corridor sites. However, with the correct vegetation regrowth parameterization, our model can capture the beetle-vegetaton-water feedbacks and could be tested in the proposed sites. By combing with fieldwork data, our model framework can help understand the dynamic changes of vegetation and hydrology after disturbances to better evaluate the water-saving efficiency of biocontrol programs.

**References**

Nagler, Pamela L., Uyen Nguyen, Heather L. Bateman, Christopher J. Jarchow, Edward P. Glenn, William J. Waugh, and Charles van Riper. 2018. "Northern Tamarisk Beetle (Diorhabda Carinulata) and Tamarisk (Tamarix Spp.) Interactions in the Colorado River Basin." *Restoration Ecology* 26 (2): 348–59. https://doi.org/10.1111/rec.12575.

Snyder, Keirith A., Russell L. Scott, and Kenneth McGwire. 2012. "Multiple Year Effects of a Biological Control Agent (Diorhabda Carinulata) on Tamarix (Saltcedar) Ecosystem Exchanges of Carbon Dioxide and Water." *Agricultural and Forest Meteorology* 164 (October): 161–69. https://doi.org/10.1016/j.agrformet.2012.03.004.

Tsamir, Mor, Sagi Gottlieb, Yakir Preisler, Eyal Rotenberg, Fyodor Tatarinov, Dan Yakir, Christina Tague, and Tamir Klein. 2019. "Stand Density Effects on Carbon and Water Fluxes in a Semi-Arid Forest, from Leaf to Stand-Scale." *Forest Ecology and Management* 453 (December): 117573. https://doi.org/10.1016/j.foreco.2019.117573.

**Author's Response to Editor**

Thank you for the helpful feedback and comments and we apologize for accidentally omitting responses to the editor's comments in our previous documents. Below are our responses to the editor's comments (our response is in red text and the editor's comments are in black text).

**Editor's comments**

(1) Thank you for submitting to the special issue. I have read the manuscript and think that it can go to the discussion forum without changes. I would have questions about the model and the modeling of water flow in the soils. I would also like to know if there is any validation of the model. But I will leave such questions to the reviewers.

The basic soil hydrologic model for RHESSys is described in detail in Tague and Band (2004) and updates described in other papers. We will provide a brief synopsis below. We are also planning to include some of this information in the supplementary material.

In RHESSys, vertical and lateral soil moisture fluxes are modeled at the patch scale (i.e., the smallest grid cell), and the connectivity between patches is organized at the subbasin scale (meaning there is a closed water budget for each subbasin in the larger watershed). RHESSys uses a 4-layer model for vertical soil moisture processes, including a surface detention store, a root accessible store, an unsaturated store below rooting depths, and a saturated store. The vertical processes also include snowpack and litter moisture stores. All vegetation layers and a litter layer can also store water through interception.

In RHESSys, rain throughfall from multiple canopy layers and a litter layer provide potential infiltration. If precipitation is snow, snow throughfall updates a snowpack store. A simplified energy budget model is used to compute snowmelt. Surface detention storage receives water through net throughfall from canopy layers and snowmelt at a daily time step. Then water infiltrates into the soil following the Phillip (1957) infiltration equation. Within the daily time step, the ponded water that is not infiltrated is added to detention storage, and any water that is above detention storage capacity generates overland flow.

Infiltration updates one of three possible stores: a saturated store in cases where water table is at the surface, a rooting zone storage, or an unsaturated store for unvegetated patches. A portion of infiltrated water is assumed to bypass the rooting zone and unsaturated store via macropores. This bypass flow directly updates a hillslope scale deeper groundwater store. Vertical drainage occurs from the unsaturated store or rooting zone store based on hydraulic conductivity. Capillary rise can move water from saturated zone to rooting zone or unsaturated store. The potential capillary rise is based on the equation from Eagleson (1978). Capillary rise is used to fill unsaturated zone to field capacity. To consider the sub-daily plant responses, 50% of capillary rise is allocated to the unsaturated zone at the beginning of the day. The rest of potential capillary rise is used to supply plant transpiration demands at the end of that day. Evaporation is computed from surface detention, surface soil and interception stores and transpiration from rooting zone or, in some cases saturated stores, using a Penman-Monteith approach.

The saturated store is modelled as a saturation deficit. Lateral fluxes occur via subsurface flow between patches or via a deeper hillslope scale groundwater flow model. Subsurface flow between patches follows topography and varies with saturation deficit and transmissivity. Transmissivity is computed as follows.

A vertical hydraulic conductivity profile is used to compute both vertical and lateral soil moisture fluxes. The saturated hydraulic conductivity, $K_{sat}(z)$ is calculated as

$$K_{sat}(z) = K_{sat_0} \, exp^{(\frac{-z}{m})} \tag{1}$$

$K_{sat_0}$: hydraulic conductivity at the surface

$m$: the decay rate of conductivity with depth

$z$: depth

Due to uncertainty in measured conductivity profiles and preferential flow, we need to calibrate $m$ and $K_{sat_0}$ against observed streamflow values. Soil porosity - $\emptyset(z)$ also changes with depth using the following equation:

$$\emptyset(z) = \emptyset_0 \, exp^{\frac{-z}{p}} \tag{2}$$

$\emptyset_0$: surface porosity which is a soil specific parameter

$p$: decay of porosity with depth

At a given profile section, the saturated soil moisture storage is computed by integrating porosity over the corresponding depth.

The drainage from the unsaturated zone to the saturated zone is controlled by two factors: field capacity of the unsaturated zone, and the vertical unsaturated hydraulic conductivity at the boundary separating the two layers. The relative saturation at field capacity is integrated over the porosity profile (from the surface to water table depth) to calculate the unsaturated zone soil moisture depth at field capacity. For this paper, the Clapp and Hornberger (1978) pedo-transfer model was used to determine the relative saturation at field capacity. Deeper groundwater flow is modelled as a simple linear aquifer.

(2) I would also like to know if there is any validation of the model.

The hydrologic sub-model is calibrated and validated against streamflow data. For the calibration period, the NSE is 0.76 with a percent error of 2.66%, and for the validation period, the NSE is 0.71 with a percent error of 8.62%. We note that RHESSys hydrologic model has been assessed in other papers in the Western US and other semi-arid systems by comparison with streamflow (e.g., Tague and Grant 2004; Garcia et al. 2013; Garcia and Tague 2015; Tague and Moritz 2019; Hanan et al. 2017), flux-tower data and sap-flow measurement (e.g., Bart et al. 2016; Tsamir et al. 2019).

For the coupled beetle effect model, we compared the model carbon and nitrogen dynamics with the result of Edburg et al. (2011) and found a strong match between modeled and observed carbon and nitrogen dynamics after beetle outbreak (eg., in Fig 5 the dynamics of the snag pool and *Live LAI* ).

For modeling the hydrological response to beetle outbreaks, we could not validate our result again observations in our study site due to lack of detailed information of mortality level and a mismatch between the timing of beetle outbreak data (i.e., from 2003 to 2012) and streamflow data (i.e., from 2011). However, we were able to compare the simulated snow sublimation changes to other recent studies. In Trail Creek, snow sublimation accounts for around 50% of total evaporation and 60% of it is canopy sublimation. The canopy sublimation accounts for an even larger proportion of total sublimation during high snow years (Fig. S7). These results are similar to other Western US forests found in recent studies (Molotch et al. 2007; Frank et al. 2019). For the most severe mortality scenario (60% evergreen mortality), we found the canopy sublimation decreases by approximately 10% when the dead foliage falls to the ground. Other process-based snow models also find similar canopy snow sublimation responses. For example, Sexstone et al. (2018) found canopy sublimation can decrease by 7% following beetle attack. Thus, several of our model results are corroborated by other studies.

**References:**

Bart, Ryan R., Christina L. Tague, and Max A. Moritz. 2016. "Effect of Tree-to-Shrub Type Conversion in Lower Montane Forests of the Sierra Nevada (USA) on Streamflow." Edited by Julia A. Jones. *PLOS ONE* 11 (8): e0161805. https://doi.org/10.1371/journal.pone.0161805.

Edburg, Steven L., Jeffrey A. Hicke, David M. Lawrence, and Peter E. Thornton. 2011. "Simulating Coupled Carbon and Nitrogen Dynamics Following Mountain Pine Beetle Outbreaks in the Western United States." *Journal of Geophysical Research: Biogeosciences* 116 (G4): G04033. https://doi.org/10.1029/2011JG001786.

Frank, John M., William J. Massman, Brent E. Ewers, and David G. Williams. 2019. "Bayesian Analyses of 17 Winters of Water Vapor Fluxes Show Bark Beetles Reduce Sublimation." *Water Resources Research* 55 (2): 1598–1623. https://doi.org/10.1029/2018WR023054.

Garcia, E. S., and C. L. Tague. 2015. "Subsurface Storage Capacity Influences Climate–Evapotranspiration Interactions in Three Western United States Catchments." *Hydrology and Earth System Sciences* 19 (12): 4845–58. https://doi.org/10.5194/hess-19-4845-2015.

Garcia, Elizabeth S., Christina L. Tague, and Janet S. Choate. 2013. "Influence of Spatial Temperature Estimation Method in Ecohydrologic Modeling in the Western Oregon Cascades." *Water Resources Research* 49 (3): 1611–24. https://doi.org/10.1002/wrcr.20140.

Hanan, Erin J., Christina (Naomi) Tague, and Joshua P. Schimel. 2017. "Nitrogen Cycling and Export in California Chaparral: The Role of Climate in Shaping Ecosystem Responses to Fire." *Ecological Monographs* 87 (1): 76–90. https://doi.org/10.1002/ecm.1234.

Molotch, Noah P., Peter D. Blanken, Mark W. Williams, Andrew A. Turnipseed, Russell K. Monson, and Steven A. Margulis. 2007. "Estimating Sublimation of Intercepted and Sub-Canopy Snow Using Eddy Covariance Systems." *Hydrological Processes* 21 (12): 1567–75. https://doi.org/10.1002/hyp.6719.

Sexstone, Graham A., David W. Clow, Steven R. Fassnacht, Glen E. Liston, Christopher A. Hiemstra, John F. Knowles, and Colin A. Penn. 2018. "Snow Sublimation in Mountain Environments and Its Sensitivity to Forest Disturbance and Climate Warming." *Water Resources Research* 54 (2): 1191–1211. https://doi.org/10.1002/2017WR021172.

Tague, C. L., and L. E. Band. 2004. "RHESSys: Regional Hydro-Ecologic Simulation System—An Object-Oriented Approach to Spatially Distributed Modeling of Carbon, Water, and Nutrient Cycling." *Earth Interactions* 8 (19): 1–42. https://doi.org/10.1175/1087-3562(2004)8<1:RRHSSO>2.0.CO;2.

Tague, Christina, and Gordon E. Grant. 2004. "A Geological Framework for Interpreting the Low-Flow Regimes of Cascade Streams, Willamette River Basin, Oregon." *Water Resources Research* 40 (4). https://doi.org/10.1029/2003WR002629.

Tague, Christina L., and Max A. Moritz. 2019. "Plant Accessible Water Storage Capacity and Tree-Scale Root Interactions Determine How Forest Density Reductions Alter Forest Water Use and Productivity." *Frontiers in Forests and Global Change* 2. https://doi.org/10.3389/ffgc.2019.00036.

Tsamir, Mor, Sagi Gottlieb, Yakir Preisler, Eyal Rotenberg, Fyodor Tatarinov, Dan Yakir, Christina Tague, and Tamir Klein. 2019. "Stand Density Effects on Carbon and Water Fluxes in a Semi-Arid Forest, from Leaf to Stand-Scale." *Forest Ecology and Management* 453 (December): 117573. https://doi.org/10.1016/j.foreco.2019.117573.

Phillip, J., 1957: The theory of infiltration: 4. Sorptivity and algebraic infiltration equation. Soil Sci., 84, 257–264.

Clapp, R., and G. Hornberger, 1978: Empirical equations for some soil hydraulic properties. Water. Resour. Res., 14, 601–604.

Eagleson, P., 1978: Climate, soil and vegetation. 3. A simplified model of soil moisture movement in the liquid phase. Water Resour. Res., 14, 722–730.